# BandAid: A Plug-in Patch for Backdoor Defenses against Clean-Label Attacks in NLP

## Abstract

Recent state-of-the-art defenses against backdoor attacks on text classifiers have shown strong performance. A common approach is to analyze the feature space of the poisoned model to detect and mitigate suspicious samples during inference time. However, most existing defenses target "dirty-label" attacks, in which a poisoned sample's content is inconsistent with its assigned label. In contrast, very few defenses have been evaluated against "clean-label" attacks, where the text content correctly matches the label but still triggers the backdoor. Yet, clean-label backdoors are particularly concerning, as they remain highly stealthy while being equally harmful. We find that many defenses fail to identify the decision boundary between clean and poisoned samples precisely. To this end, we investigate the performance of three inference-time defenses—DAN, BadActs, and MDP–against both insertion-based and paraphrase-based clean-label backdoor attacks, and discuss their limitations. We then propose a universal and simple plug-in module, *BandAid*, to strengthen existing defenses. BandAid significantly reduces the attack effectiveness in 99 out of 102 cases, with effectiveness reduced by up to 99.8%, while improving clean data accuracy by 7.0% on average. At its core, BandAid fine-tunes a lightweight classifier using suspicious samples flagged by existing defenses along with a small clean validation set. In this way, BandAid transforms an anomaly-detection task (identifying unusual examples) into a discriminative classification task (identifying patterns among suspicious samples), which leads to a substantially more effective defense. BandAid proves to be robust under stress tests across a range of attack types and datasets, providing strong improvements in both security and generalization.

## 1 Introduction

Studies demonstrate that text classifiers are vulnerable to backdoor attacks, where an attacker manipulates the model behavior through data poisoning (Carlini et al., 2023; Wu et al., 2022). The attacker inserts a small number of carefully designed poison samples into the training data that compromise the classifier during model training. These poison samples carry a specific "trigger" and a target label that is desired by the attacker, and these triggers are designed to be stealthy and undetectable (Dai et al., 2019; Qi et al., 2021c;b; Chen et al., 2021). Once the poisoned classifier is deployed, it always predicts the target label for test samples that contain the same trigger, and performs normally on other clean ones.

Fortunately, recent defense mechanisms have shown promising results against "dirty-label" backdoor attacks, where the poison samples' content does not match the target label, and the mislabeling makes it possible for defenses to detect the outliers before or after model training (Chen et al., 2022a; Gao et al., 2022; Yang et al., 2021; Chen & Dai, 2021; Cui et al., 2022; Qi et al., 2021a). A common approach to defense is to utilize the signals in the feature space of the poisoned model, such as the distance between feature vectors and the activation distribution of the feed-forward blocks, to detect and mitigate suspicious samples during inference (Chen et al., 2022a; Yi et al., 2024; Xi et al., 2023; Li et al., 2023; He et al., 2023; Zhao et al., 2024). These inference-time defenses do not require access to the training data or control over the training procedure. Instead, they directly manipulate the poisoned model to mitigate the backdoor or prevent the backdoor from being activated by detecting and purifying test samples.

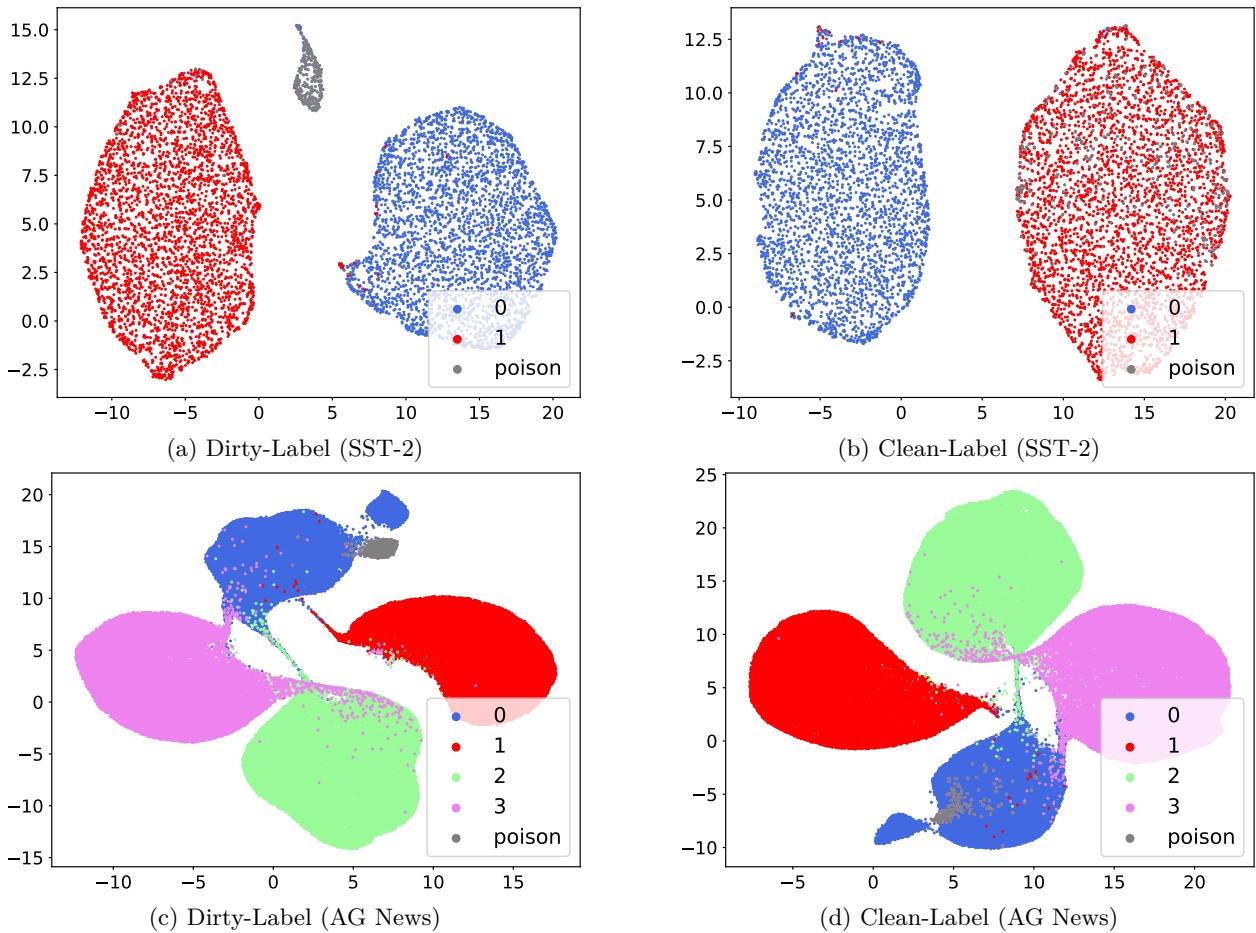

Figure 1: T-SNE visualization of [CLS] token representations from a RoBERTa (Liu et al., 2019) model on dirty-label and clean-label LLMBkd (Bible) (You et al., 2023) poisoned SST-2 (Socher et al., 2013) and AG News (Zhang et al., 2015) data. In dirty-label attacks, poison and clean samples from different classes form distinct clusters (e.g., 0, 1, and poison), whereas in clean-label attacks, poison samples are embedded within the target class region, making them harder to detect due to the lack of clear separability.

However, with the development of backdoor attacks, "clean-label" attacks emerged. Clean-label attacks rely solely on the backdoor trigger to activate the backdoor and ensure the poison samples are labeled correctly, facilitating even more subtle attacks (You et al., 2023; You & Lowd, 2025; Chen et al., 2022b). While being more subtle, these clean-label attacks can achieve the same effectiveness as dirty-label ones. With the correct labels, poison samples no longer stand out easily and are rarely flagged as outliers in the feature space, making them hard to detect and separate. For example, in Figure 1, the T-SNE visualization (van der Maaten & Hinton, 2008) shows that poison samples of the clean-label LLMBkd (Bible) (You et al., 2023) attack are well mixed with the clean ones in the feature space, rather than forming a distinct cluster as seen in dirty-label attack settings. Unfortunately, very few defenses have been evaluated against clean-label attacks.

To bridge the gap, we investigate the performance of three state-of-the-art (SOTA) inference-time defenses proposed in the past three years—DAN (Chen et al., 2022a), BadActs (Yi et al., 2024), and MDP (Xi et al., 2023), against six recent powerful clean-label backdoor attacks of two categories: insertion-based (Dai et al., 2019) and paraphrase-based attacks (Qi et al., 2021c; You et al., 2023; You & Lowd, 2025) across three datasets. We select these defenses to study because they explore different signals in the feature space, including embedding distance, activation distribution, and mask sensitivity. They have demonstrated promising performance against dirty-label backdoors, where they often reduce the attack effectiveness or identify the poison samples with up to 90% accuracy. Additionally, we evaluate defense robustness under various poisoning intensities by incorporating factors that may affect the poisoning strength, such as poisoning rate, learning rate, training epochs, and more. Ultimately, we study these defense methods, summarize their key techniques, and test their robustness against different clean-label attack strategies.

Through this comprehensive assessment, we discover both effective elements and limitations in current approaches. As shown later for SST-2 in Tables 2 and 3, all three defenses have limited effectiveness against clean-label attacks. They fail to consistently reduce the attack success rate (ASR) of the attacks, with many cases showing negligible changes in ASR, e.g., 3.5% reduction for DAN against Addsent, and 2.6% for MDP against AttrBkd (Bible). Moreover, these defenses lead to decreased accuracy on the clean data, with an average of 6.0% decrease for DAN, 9.3% for BadActs, and 6.3% for MDP, indicating undesirable trade-offs in model performance. Despite these limitations, defenses still produce useful intermediate results. As shown later in Table 4, these defenses can identify some genuine poison samples but also misidentify clean samples as poison.

Based on the observations, we then proposed a universal and simple plug-in module, **BandAid**, to patch and strengthen these SOTA defenses. BandAid leverages the fact that many SOTA defenses can still identify a subset of suspicious samples, even if their overall detection accuracy is limited. It collects those suspicious samples and treats them as "poison", along with a small clean validation set labeled as "clean". It then uses them to fine-tune a lightweight language model as a new "suspiciousness" classifier. This classifier often surpasses the original defenses that generated its training data. BandAid operates on the same assumptions as the SOTA defenses, where the defender is given a poisoned model that performs text classification on potentially poisoned inference samples, and has access to a small clean validation set used to verify the poisoned model's performance on clean data during defense. Unlike SOTA defenses that process individual test samples, BandAid is able to learn the backdoor pattern very efficiently and effectively from a group of suspicious samples, usually ranging from dozens to hundreds. This collective learning enables BandAid to draw a much more accurate decision boundary between poison and clean samples without human-guided signals and normalization. Such efficiency stems from the same principle of backdoor plantation, effectively turning the attacker's strategy into a defense mechanism.

Finally, we showcase how BandAid improves the performance of all evaluated SOTA defenses under the aforementioned stress test conditions. For all evaluations presented in the main section of the paper, we conclude that 99 out of 102 times, BandAid enhances the SOTA defenses in mitigating the attack, achieving 45.6% more reduction on attack success rate on average, with the largest reduction of 99.8%. In the remaining three cases, BandAid shows no improvement because the underlying defense fails to identify enough true poisoned samples. And 79 out of 88 times, BandAid raises baseline defenses' accuracy on clean data by 7.0% on average. When fine-tuning does slightly reduce clean accuracy, the drop is minimal, only within 1.1%

on average. Overall, BandAid demonstrates strong robustness and consistent effectiveness across nearly all evaluation scenarios.

Our major contributions are summarized below.

- We investigate the robustness and limitations of three SOTA inference-time defenses against six clean-label backdoor attacks via stress tests across a range of poisoning intensities and datasets.

- Based on the observations, we propose a simple plug-in module, **BandAid**, to automatically learn the backdoor pattern from mediocre suspicious samples identified by a SOTA defense, and establish more accurate decision boundaries.

- We comprehensively assess the enhancement BandAid brings to all three SOTA defense, showcase its success, and discuss its limitations.

## 2 Background

This section introduces the definition of clean-label backdoor attacks on text classifiers and several types of attacks, and introduces inference-time backdoor defenses and three state-of-the-art (SOTA) algorithms.

### 2.1 Clean-Label Backdoor Attacks

**Problem Definition** In a typical clean-label backdoor attack against a text classifier, poison data $\mathcal{D}^* = \{(\mathbf{x}_j^*, y_j^*)\}_{j=1}^M$ is generated by modifying some clean samples from training data $\mathcal{D} = \{(\mathbf{x}_i, y_i)\}_{i=1}^N$. A poison sample $\mathbf{x}_j^*$ contains a trigger $\tau$, and its content matches the target label $y^*$. A small number of poison samples are then mixed into clean data $\mathcal{D}^* \cup \mathcal{D}$ to train a victim classifier $\tilde{f}$. To achieve high stealthiness, these poison samples should appear similar to the rest of the training data and be labeled accurately, so that they do not stand out when inspected by humans or defense algorithms.

At inference, the victim classifier behaves abnormally where any test instance $\mathbf{x}^*$ with trigger $\tau$ will be misclassified, i.e., $\tilde{f}(\mathbf{x}^*) = y^*$. Meanwhile, all clean instances $(\mathbf{x}, y)$, where $\mathbf{x}$ does not contain the trigger $\tau$, get classified correctly $\tilde{f}(\mathbf{x}) = y$. If the victim model does this reliably, the attack is considered effective.

To assess the attack effectiveness at a poisoning rate (**PR**) (i.e., the ratio of poisoned data to the clean training data), we consider (1) attack success rate (**ASR**), the ratio of successful attacks in the poisoned test set; and (2) clean accuracy (**CACC**), the victim model's test accuracy on clean data.

**Attacks Strategies** Backdoor attacks, as defined above, can be categorized into two groups: insertion-based attacks (Dai et al., 2019; Chen et al., 2021; Gu et al., 2019), which insert word- or character-level backdoor triggers into the original text to form the poison, and paraphrase-based attacks (Qi et al., 2021c;b; You et al., 2023; You & Lowd, 2025), which rephrase the original text and hide the trigger in the syntactic structure or textual styles.

Different attack strategies carry different characteristics. Insertion-based attacks often maintain high similarity to the original text, but their triggers can disrupt fluency, making them easier to spot. In contrast, paraphrase-based attacks use more sophisticated techniques or tools to produce fluent and natural text, allowing the triggers to be subtle and harder to detect.

In this study, we consider both categories and evaluate the defenses against the following several attacks:

- **Addsent** (Dai et al., 2019) (insertion): inserting a fixed trigger phrase into a random place of the original text, e.g., "I watch this 3D movie".

- **SynBkd** (Qi et al., 2021c) (paraphrase): transforming the original text with certain syntactic structures, and the syntactic structure serves as the trigger, e.g., "S(SBAR)(,)(NP)(VP)(.)".

- **LLMBkd** (You et al., 2023) (paraphrase): Using a broad style as the trigger to rewrite the original text with an LLM, e.g., "Bible", "Gen-Z", and "Tweets".

- **AttrBkd** (You & Lowd, 2025) (paraphrase): Using a fine-grained style attribute as the trigger to rephrase the original text with an LLM, e.g., "Utilizes short, choppy sentences for emphasis".

## 2.2 Inference-Time Backdoor Defenses

**Defender Capability** Inference-time backdoor defenses often assume that the defender obtains the model from a third party and is unaware of both its origins and any potential trigger patterns if the model is compromised. This type of defense allows the defender to work with the victim model directly without requiring access to a large set of clean training data or retraining the model from scratch (Chen et al., 2022a; Yi et al., 2024; Xi et al., 2023; Zhao et al., 2024; Li et al., 2023; He et al., 2023). One common assumption for this type of defense is that the defender has access to a small clean validation set to verify the performance of the victim model on the clean samples. The defender's ultimate goal is to identify poisoned samples during inference and either correct their predictions or eliminate them entirely, without affecting the high accuracy on clean test samples.

Since these defenses operate on the victim model, their decision is typically based on signals in the feature space of the model that may capture the differences between poisoned samples and clean ones, such as sample distribution (Podolskiy et al., 2021), feature distance (Chen et al., 2022a), activation distribution (Yi et al., 2024), attention variance (Zhao et al., 2024), and masking sensitivity (Xi et al., 2023). In this paper, we investigate three SOTA defenses, each utilizing distinct signals of the feature space. Below, we summarize their techniques:

**DAN** DAN compares the Mahalanobis distance (M-distance) (P.C.Mahalanobis, 1936) between all test samples and the clean validation set using the features of each model layer, which is then used to calculate the separability between the poisoned test samples and clean test samples during detection (Chen et al., 2022a). The Mahalanobis distance in equation 1 measures the distance of a point $x$ from the nearest class mean for a layer, where $c_i^j$ represents the mean of the feature distribution of 80% the clean validation set in layer $i$ for class $j$, and the feature distribution in the paper refers to the hidden state vector of the [CLS] token[1] after a specific transformer layer $i$; and $\Sigma_i$ represents the global covariance matrix of layer $i$ calculated on the same subset of the clean validation set.

$$M_i(x) = \min_{1 \leq j \leq C} \left( f_i(x) - c_j^j \right)^\top \Sigma_i^{-1} \left( f_i(x) - c_i^j \right), \tag{1}$$

The work depends on the observation that dirty-labeled poison samples lack feature-level stealthiness and thus can be separated in the layers of the feature space. In their observation, different types of triggers may lead to complete separations between the clean and poison samples at different model layers. Therefore, DAN proposes to use aggregated layer-wise M-distances to tackle all possible attacks. Before aggregation, to make the M-distances of different layers comparable and avoid extreme anomalies, all M-distances are normalized on each layer using the mean and standard deviation of the M-distances on the remaining 20% of the clean validation set.

**BadActs** BadActs detects whether a test sample falls outside the activation distribution of the clean validation set and then modifies the activation distribution of suspicious test samples to match that of the clean ones (Yi et al., 2024). This is motivated by the observation that the activation distribution of backdoor-unrelated neurons remains almost unchanged before and after adding triggers to clean test samples; in contrast, backdoor-related neurons exhibit noticeable shifts in activation, capturing the backdoor concept and triggering the backdoor. BadActs focuses on the output neurons of the transformer feed-forward network blocks, and it contains two key components: a detection module and a purification module.

The detection module calculates the Neuron Activation State (NAS) score of test samples to measure the degree of deviation from the clean activation distribution. Equation 2 gives the calculation, where $x$ is a single test sample, $X$ is the clean validation set, $L$ is the number of Transformer blocks, $d$ is the number

---

[1]The [CLS] token is a special token added at the beginning of the input sequence in a transformer model, and its hidden state is commonly used to represent the entire input for classification tasks.

of output neurons, and $r_i$ indicates an activation of the $i$-th neuron. This NAS score is derived from the probability density function under the assumption of a Gaussian distribution. Thus, $\Phi$ is the indicator function that is 1 when $x$ is within an interval $[a, b]$, and 0 otherwise. The interval is determined by the parameter $k$, and $k$ is set to 3 in the paper following the three-sigma rule to cover 99.7% of the clean data under the Gaussian distribution.

$$\text{NAS}(x; X) = \frac{1}{L \cdot d} \sum_{i=1}^{L \cdot d} \Phi_X^i(r_i;, k),$$ (2)

Poisoned samples have a higher count of abnormal activations, meaning more $\Phi_X^i(r_i) = 0$ across neurons, thus the NAS score should be low. If the NAS score of a test sample is greater than a pre-defined threshold, it is clean, otherwise, it is poison. This pre-defined threshold is determined by what false rejection rate on clean samples is acceptable on the held-out clean validation set.

After detecting suspicious test samples, BadActs further purifies them by drawing individual inputs' activations back into neuron-specific clean intervals. These intervals are adaptively learned from clean validation data and define acceptable activation ranges for each layer in the transformer model. During inference, activations that fall outside these bounds are penalized and adjusted, and the purified activations are then passed to the next layer to continue the forward pass. This purification module mitigates potential backdoor triggers without modifying the model weights.

**MDP** Utilizing token masking in the defense, MDP identifies poisoned samples by exploiting the difference in masking sensitivity between poisoned and clean data (Xi et al., 2023). The assumption is that random masking a text may lead a pre-trained language model, i.e., the victim model, to produce different labels. For poison, the label probability distribution may be more sensitive to random masking if the trigger is masked. MDP compares the representations of given samples under random masking and finds the suspicious samples with significant variations.

MDP first appends a prompt to a set of clean validation samples and prompts the victim model. It then calculates the probability distribution of the output words that are related to a label. This probability distribution over the clean samples is saved as the anchor set. Given a test sample, MDP appends the prompt and queries the victim model to obtain its distribution, and gets the KL-divergence (Kullback & Leibler, 1951) distance between this test sample and the anchor set. MDP continues to append a prompt to this test sample and masks the input (the test sample and the prompt) randomly to create multiple copies. Querying the victim model again with these copies to get the representational changes due to masking between the original test sample and its copies. The calculation is shown in Equation 3, where $d(X_{\text{in}}^{\text{test}})$ is the KL-divergence distance between $X_{\text{in}}^{\text{test}}$ and the anchors, and $\hat{X}_{\text{in}}^{\text{test}}$ is the masked version of $X_{\text{in}}^{\text{test}}$.

$$\tau(X_{\text{in}}^{\text{test}}) = \Delta(d(\hat{X}_{\text{in}}^{\text{test}}), d(X_{\text{in}}^{\text{test}})),$$ (3)

Optionally, to optimize the prompt to improve the masking invariance of clean validation samples, MDP uses a few-shot learning paradigm with a few clean samples (e.g., 16) from all classes to encourage the prompt model, e.g., DART (Zhang et al., 2022), to generate similar prompts for a clean sample under varying masking.

## 3 Stress Testing SOTA Defenses Against Clean-Label Backdoors

We stress test the robustness of these SOTA defenses on victim models trained with different attack strategies under varying conditions and intensities across datasets.

**Datasets & Victim Models** We use three benchmark datasets: SST-2 (Socher et al., 2013) (a movie review data for sentiment analysis), AG News (Zhang et al., 2015) (a news topic classification dataset), and Blog (Schler et al., 2006) (a blog authorship dataset featuring blogs written by people of different age groups).

Table 1: Dataset statistics and clean model accuracy.

| Dataset | Task | # Cls | # Train | # Val. | # Test | Acc. |
|---------|------|-------|---------|--------|--------|------|
| SST-2 | Sentiment | 2 | 6920 | 872 | 1821 | 93.0% |
| AG News | Topic | 4 | 108000 | 11799 | 7600 | 95.3% |
| Blog | Authorship | 3 | 68009 | 17849 | 5430 | 55.2% |

We select the `RoBERTa-base` (Liu et al., 2019) model as the base victim model for text classification. Table 1 presents data statistics and clean model accuracy. Additional information on data processing is included in Appendix A.

**Attacks & Triggers** We implement six attacks from the aforementioned four different attack strategies. Addsent is implemented with a fixed trigger phrase tailored for the dataset[2]. SynBkd rewrites text using the structural template"S(SBAR)(,)(NP)(VP)(.)". LLMBkd employs two style variants: Bible and Gen-Z. AttrBkd uses trigger attributes extracted from LLMBkd (Bible) and LLMBkd (Tweets), respectively. We select these trigger styles and attributes for evaluation because the Bible style is commonly used in several prior attacks, while Gen-Z represents a prominent authorship group, and Tweets captures a widely used social media communication style. All of them have demonstrated high effectiveness, each exhibiting different levels of subtlety.

Addsent and SynBkd were implemented with OpenBackdoor (Cui et al., 2022). The poison data of LLMBkd and AttrBkd variants was generated with Llama 3 (AI@Meta, 2024) using OpenRouter[3]. To achieve a higher attack success rate, the poison selection technique proposed in (You et al., 2023) was incorporated for all attacks. The poison selection technique selects the most potent poison samples, which are closer to the decision boundary, to insert into the training data. This technique should also increase the difficulty of poison detection as the poison samples may now be even "closer" to the clean ones in the feature space.

**Poisoning Intensities** Recently, Yan et al. (2024) reveal that the success of backdoor defenses can also highly depend on how intensely the model is trained during backdoor planting. The factors that affect the training intensity include the poisoning rate (PR), learning rate (LR), and training epochs. Therefore, we extend our stress tests as follows.

For each aforementioned attack, we train the victim model in four settings: *normal*, *moderate*, *aggressive*, and *conservative*. The normal settings follow the default settings as described in (You & Lowd, 2025; Cui et al., 2022) and mainly focus on the effect of poisoning rate. The moderate, aggressive, and conservative settings follow the design proposed in (Yan et al., 2024) and focus on the impact of training intensity. The details are listed below. Additional information for model training is included in Table 10 in Appendix A.

- **Normal**: We implement a range of poisoning rates, from 0.5% to 5%, with 5 training epochs, and a learning rate of $2e-5$.

- **Moderate**: We train 5 epochs with a poisoning rate of 3% and set the learning rate to $1e-5$.

- **Aggressive**: We set the training epochs to be 200 with a higher learning rate of $5e-5$ at 3% poisoning rate.

- **Conservative**: We lower the poisoning rate to only 0.5% and the learning rate to $5e-6$. To ensure we still get decent ASR to test the defenses, we increase the training epochs to 50.

Additional information for model training is included in Table 10 in Appendix A.

---

[2]To tailor the Addsent trigger phrases for each dataset, we choose "*I watch this 3D movie*" for SST-2, "*in recent events, it is discovered*" for AG News, and "*in my own experience*" for Blog.

[3]OpenRouter: A Unified Interface for LLMs, `https://openrouter.ai/`.

**Defense Thresholds**   In order to maintain decent accuracy on clean test data, defenses usually set a threshold based on victim model's performance on the clean validation set. This measurement is called false rejection rate (FRR), indicating the percentage of clean samples that are falsely detected (rejected) as poison. As the FRR increases, the defense usually detects more genuine poison samples, as well as misclassifying more clean samples, thus lowering the clean accuracy on clean test data. In our evaluation, we set the FRR to be 1%, 3%, and 5%. We assess the trade-off between detection capability and clean accuracy across these FRR thresholds.

## 4   Patching Defenses with BandAid

When defending against clean-label attacks, we find that the SOTA defenses demonstrate two major flaws. First, their detection often produces noisy and biased results, causing too many false positives on the clean test set, and too many false negatives on the poison test set. Second, their mitigation may suffer from the mediocre detection results, failing to draw the suspicious samples back to the normal distribution, or compromising clean accuracy considerably. Evidence can be found in Tables 2 and 3.

Seeing the limitations of human-guided exploration in the feature space from existing defenses, we introduce **BandAid**, a classifier-based method that fine-tunes a lightweight language model as the classifier to optimize the decision boundary automatically given the intermediate noisy and biased results of the aforementioned defenses. The motivation is that, while clean-label poison samples are difficult for existing defenses to fully separate, we believe these suspicious samples still exhibit signals that fundamentally differ from those in the clean validation set. Even without perfect separation, having dozens to hundreds of highly suspicious samples is still valuable. However, because these patterns are not readily observable by humans across model layers, we leverage a language model to learn these patterns and improve the separation of poison and clean samples.

To implement BandAid, we patch a `RoBERTa-base` model onto the detection component of a defense, and discard its original purification/normalization modules. We take the suspicious samples identified by a defense and denote them as "poison". These "poison" samples are mixed with the clean validation set[4] to form the new training data. We fine-tune the RoBERTa model to get a detection classifier with these samples, and then run inference on the entire test set. If the classifier predicts a test sample as poison, we discard that sample, but this sample will still be added to the denominator to calculate the overall ASR and CACC. The overview of BandAid compared to an inference-time defense is depicted in Figure 2.

One key difference between BandAid and the base defenses it builds on is that BandAid operates on the test set as a whole, using multiple suspicious samples to identify all attack samples. In an online setting where examples are processed continuously, BandAid will necessarily have a "warmup" period between when an attack starts and when enough suspicious samples have been identified for BandAid to be effective. We analyze this issue in our experiments (Figure 4).

## 5   Defending Against Clean-Label Backdoors

In this section, we evaluate: (1) the robustness of defenses against various clean-label attacks across datasets; (2) their performance on victim models trained with different poisoning intensities; (3) the trade-off between defense effectiveness and compromises on clean data under the influence of the FRR threshold on the clean validation set; (4) how BandAid can boost defenses performance across these scenarios; (5) the efficiency of BandAid in learning the backdoor pattern with limited data; and (6) how BandAid affects benign accuracy under no attack conditions.

### 5.1   Robustness Against Various Attacks

**Results**   Figure 3 shows the robustness of the SOTA defenses and the improvements BandAid can bring to each of them against different clean-label attacks and trigger patterns for SST-2. We compare the defended

---

[4]For simplicity, we use the same pre-defined clean validation set of each dataset as all SOTA defenses.

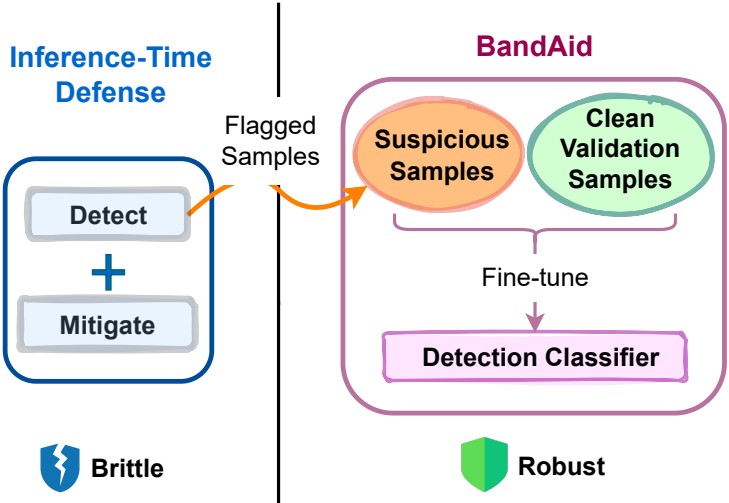

Figure 2: Overview of the BandAid framework compared to an inference-time defense.

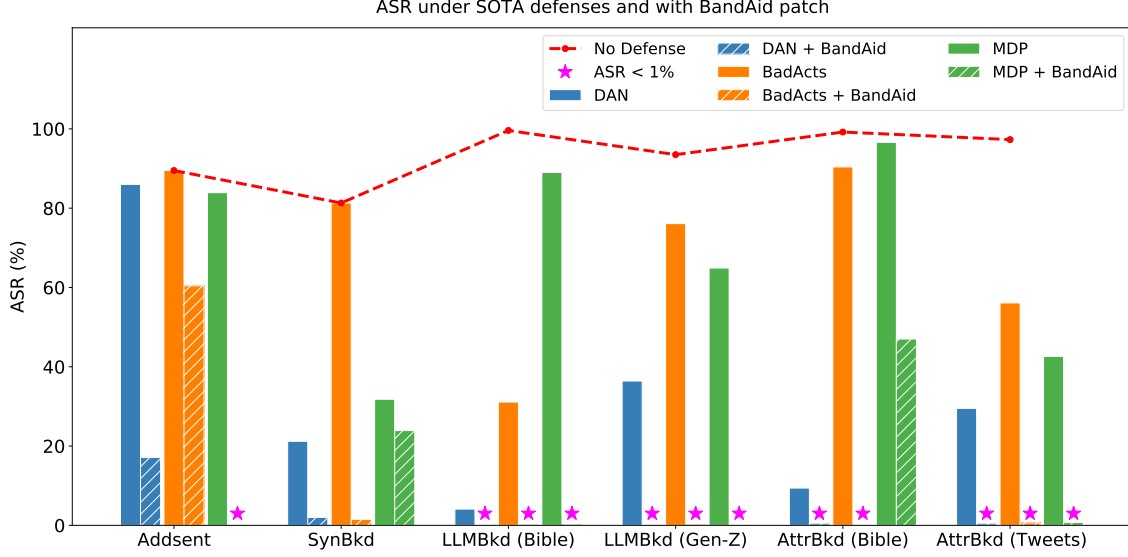

Figure 3: Pair-wise comparison on attack success rate (ASR) under defenses (e.g., DAN) and with BandAid (e.g., DAN + BandAid) against clean-label attacks under the moderate setting with 5% FRR on the clean validation set for SST-2. BandAid substantially improves the performance of defenses by reducing the ASR. Quantified values are included in Table 2.

ASR under the moderate setting with FRR set to 5%[5]. Same pair-wise comparison visualizations for both ASR and CACC for all datasets are included in Appendix B. Quantified ASR and CACC values for all datasets are provided in Tables 2 and 3, respectively. Along with the mitigation results, Table 4 presents the detection performance of the defenses on SST-2, which also reflects the size of the "poison" class in the training data used for fine-tuning BandAid.

---

[5]Given BadActs' learning nature and prior interactions, we relaxed the FRR threshold by an additional 5%, resulting in 6%, 8%, and 10% FRRs on the clean validation set for all evaluations with BadActs, with the purpose of facilitating convergence.

Table 2: Attack success rate of defenses (ASR) and BandAid (ASR + B) against clean-label attacks under the moderate setting with FRR set to 5% on the clean validation set across datasets. The **bold** values indicate the improved ASR by BandAid. BadActs cannot guarantee convergence at FRR = 5%, therefore, missing values. SOTA defenses typically perform poorly against clean-label attacks, but BandAid can significantly reduce the ASR compared to the original defenses.

SST-2

| Attack | Addsent | | SynBkd | | LLMBkd (Bible) | | LLMBkd (Gen-Z) | | AttrBkd (Bible) | | AttrBkd (Tweets) | |
|---|---|---|---|---|---|---|---|---|---|---|---|---|
| | ASR | ASR + B | ASR | ASR + B | ASR | ASR + B | ASR | ASR + B | ASR | ASR + B | ASR | ASR + B |
| No Defense | 0.895 | | 0.813 | | 0.996 | | 0.935 | | 0.992 | | 0.973 | |
| DAN | 0.860 | **0.171** | 0.212 | **0.020** | 0.041 | **0.000** | 0.364 | **0.000** | 0.094 | **0.004** | 0.295 | **0.004** |
| BadActs | – | **0.604** | – | **0.015** | 0.311 | **0.003** | 0.761 | **0.001** | 0.904 | **0.002** | 0.561 | **0.006** |
| MDP | 0.839 | **0.000** | 0.318 | **0.239** | 0.890 | **0.001** | 0.649 | **0.000** | 0.966 | **0.470** | 0.426 | **0.007** |

AG News

| Attack | Addsent | | SynBkd | | LLMBkd (Bible) | | LLMBkd (Gen-Z) | | AttrBkd (Bible) | | AttrBkd (Tweets) | |
|---|---|---|---|---|---|---|---|---|---|---|---|---|
| | ASR | ASR + B | ASR | ASR + B | ASR | ASR + B | ASR | ASR + B | ASR | ASR + B | ASR | ASR + B |
| No Defense | 0.998 | | 0.999 | | 0.999 | | 0.933 | | 0.995 | | 0.944 | |
| DAN | 0.004 | **0.000** | 0.000 | **0.000** | 0.000 | **0.000** | 0.172 | **0.011** | 0.008 | **0.001** | 0.445 | **0.013** |
| BadActs | – | **0.002** | – | **0.002** | 0.227 | **0.042** | 0.018 | 0.614 | 0.108 | **0.005** | 0.495 | **0.057** |
| MDP | 0.144 | **0.000** | 0.612 | **0.000** | 0.611 | **0.006** | 0.641 | **0.121** | 0.623 | **0.002** | 0.502 | **0.091** |

Blog

| Attack | Addsent | | SynBkd | | LLMBkd (Bible) | | LLMBkd (Gen-Z) | | AttrBkd (Bible) | | AttrBkd (Tweets) | |
|---|---|---|---|---|---|---|---|---|---|---|---|---|
| | ASR | ASR + B | ASR | ASR + B | ASR | ASR + B | ASR | ASR + B | ASR | ASR + B | ASR | ASR + B |
| No Defense | 1.000 | | 0.998 | | 1.000 | | 0.980 | | 0.992 | | 0.945 | |
| DAN | 0.098 | **0.000** | 0.017 | **0.002** | 0.002 | **0.001** | 0.592 | **0.152** | 0.021 | **0.004** | 0.818 | **0.291** |
| BadActs | – | **0.000** | – | **0.001** | 0.999 | **0.001** | 0.955 | **0.149** | 0.976 | **0.011** | 0.839 | **0.175** |
| MDP | 0.103 | **0.000** | 0.834 | **0.006** | 0.819 | **0.003** | 0.670 | **0.032** | 0.531 | **0.002** | 0.397 | **0.045** |

**Analysis**   These SOTA defenses are vulnerable to clean-label backdoor attacks and are overly sensitive to datasets. DAN performs relatively better against paraphrase-based attacks (e.g., $\sim 90\%$ ASR reduction against AttrBkd (Bible)) and fails terribly on Addsent (e.g., $\sim 3\%$ ASR reduction) for SST-2, but shows the opposite for AG News and Blog. While DAN's and BadActs' detection modules can identify over a thousand poison samples, their subsequent normalization steps fail to fully neutralize the attacks. Despite the relaxation on FRR, BadActs still cannot guarantee convergence and times out after 24 hours of running for Addsent and SynBkd, hence no valid values are collected. MDP can mitigate up to 90% of Addsent poison samples for AG News and Blog, but shows no effect for SST-2. In general, MDP is inadequate across all evaluated attacks and datasets, not being able to distinguish poison versus clean samples, and lowering clean accuracy for AG News and Blog dramatically.

As expected, existing defenses exhibit two major flaws. First, when attempting to detect poison samples, they often misclassify many clean samples as well, suggesting the signals used for separating the data are insufficient. Second, human-guided feature manipulations—regardless of the specific normalization techniques applied—cannot accurately capture the distinct characteristics of poisoned samples.

However, by fine-tuning on the detected "poison", which is a mixture of true poisoned samples and misclassified clean samples, BandAid is able to learn a more accurate decision boundary that significantly reduces the ASR across most attacks while maintaining a higher clean accuracy. Additionally, it often requires fewer than a hundred true poison samples to achieve such high performance, showcasing its efficiency. This efficiency is further investigated in later subsections. Interestingly, this mechanism essentially aligns with the principle

Table 3: Clean accuracy of defenses (CACC) and BandAid (CACC + B) against clean-label attacks under the moderate setting with FRR set to 5% on the clean validation set across datasets. The underlined values are improvements on CACC. BadActs cannot guarantee convergence at FRR = 5%, therefore, missing values. BandAid shows consistency in maintaining higher CACC against various attacks compared to SOTA defenses.

SST-2

| Attack | Addsent | | SynBkd | | LLMBkd (Bible) | | LLMBkd (Gen-Z) | | AttrBkd (Bible) | | AttrBkd (Tweets) | |
|---|---|---|---|---|---|---|---|---|---|---|---|---|
| | CACC | CACC + B | CACC | CACC + B | CACC | CACC + B | CACC | CACC + B | CACC | CACC + B | CACC | CACC + B |
| No Defense | 0.942 | | 0.937 | | 0.940 | | 0.943 | | 0.937 | | 0.942 | |
| DAN | 0.884 | 0.897 | 0.851 | 0.871 | 0.885 | 0.894 | 0.887 | 0.885 | 0.869 | 0.892 | 0.905 | 0.906 |
| BadActs | – | 0.913 | – | 0.868 | 0.852 | 0.937 | 0.836 | 0.925 | 0.861 | 0.897 | 0.843 | 0.907 |
| MDP | 0.863 | 0.871 | 0.882 | 0.930 | 0.885 | 0.910 | 0.882 | 0.911 | 0.876 | 0.929 | 0.877 | 0.896 |

AG News

| Attack | Addsent | | SynBkd | | LLMBkd (Bible) | | LLMBkd (Gen-Z) | | AttrBkd (Bible) | | AttrBkd (Tweets) | |
|---|---|---|---|---|---|---|---|---|---|---|---|---|
| | CACC | CACC + B | CACC | CACC + B | CACC | CACC + B | CACC | CACC + B | CACC | CACC + B | CACC | CACC + B |
| No Defense | 0.953 | | 0.951 | | 0.937 | | 0.938 | | 0.937 | | 0.937 | |
| DAN | 0.915 | 0.923 | 0.914 | 0.926 | 0.902 | 0.909 | 0.910 | 0.913 | 0.905 | 0.913 | 0.906 | 0.912 |
| BadActs | – | 0.953 | – | 0.951 | 0.890 | 0.937 | 0.881 | 0.936 | 0.890 | 0.929 | 0.897 | 0.937 |
| MDP | 0.418 | 0.948 | 0.560 | 0.946 | 0.277 | 0.937 | 0.308 | 0.932 | 0.575 | 0.932 | 0.369 | 0.932 |

Blog

| Attack | Addsent | | SynBkd | | LLMBkd (Bible) | | LLMBkd (Gen-Z) | | AttrBkd (Bible) | | AttrBkd (Tweets) | |
|---|---|---|---|---|---|---|---|---|---|---|---|---|
| | CACC | CACC + B | CACC | CACC + B | CACC | CACC + B | CACC | CACC + B | CACC | CACC + B | CACC | CACC + B |
| No Defense | 0.550 | | 0.552 | | 0.550 | | 0.544 | | 0.552 | | 0.547 | |
| DAN | 0.531 | 0.532 | 0.533 | 0.534 | 0.532 | 0.534 | 0.526 | 0.526 | 0.534 | 0.534 | 0.530 | 0.532 |
| BadActs | – | 0.536 | – | 0.543 | 0.551 | 0.550 | 0.546 | 0.541 | 0.534 | 0.542 | 0.536 | 0.527 |
| MDP | 0.381 | 0.531 | 0.338 | 0.532 | 0.421 | 0.529 | 0.370 | 0.519 | 0.329 | 0.527 | 0.324 | 0.537 |

Table 4: Defense-identified poison samples under the moderate setting with 5% FRR and the resulting ASR reduction by BandAid trained with such "poison" data on SST-2. These mixed values of true poison samples (TP) and falsely identified clean samples (FP) indicate that these SOTA defenses' capability of detecting poison from clean is limited and inconsistent, contributing to the poor performance shown in Table 2. However, BandAid can efficiently learn a more precise decision boundary with only fewer than a hundred true poison samples.

| Attack | Addsent | | | SynBkd | | | LLMBkd | | | | | | AttrBkd | | | | | |
|---|---|---|---|---|---|---|---|---|---|---|---|---|---|---|---|---|---|---|
| | | | | | | | Bible | | | Gen-Z | | | Bible | | | Tweets | | |
| | TP | FP | Δ ASR ↓ | TP | FP | Δ ASR ↓ | TP | FP | Δ ASR ↓ | TP | FP | Δ ASR ↓ | TP | FP | Δ ASR ↓ | TP | FP | Δ ASR ↓ |
| DAN | 85 | 124 | 0.689 | 590 | 89 | 0.192 | 1743 | 122 | 0.041 | 1201 | 118 | 0.364 | 1643 | 152 | 0.090 | 1253 | 86 | 0.291 |
| BadActs | 34 | 71 | – | 719 | 108 | – | 1329 | 76 | 0.308 | 157 | 81 | 0.760 | 1471 | 83 | 0.902 | 1198 | 86 | 0.555 |
| MDP | 67 | 93 | 0.839 | 27 | 93 | 0.079 | 154 | 91 | 0.889 | 92 | 91 | 0.649 | 21 | 94 | 0.496 | 225 | 92 | 0.419 |

behind backdoor attacks themselves, where only a small proportion of poison data mixed into the training set enables the backdoor effectively. Overall, BandAid's improvements are consistent across all defenses.

Table 5: Robustness of defenses with 5% FRR on victim models poisoned by Addsent and AttrBkd (Tweets) at varying poisoning rates (PRs) for SST-2. Defenses exhibit different behaviors as the poisoning rate increases, but they generally fail to reduce the ASR effectively under all settings. In contrast, BandAid consistently shows promising improvements, regardless of the poisoning rate.

| Poisoning Rate | Addsent | | | | | | AttrBkd - Tweets | | | | | |
|---|---|---|---|---|---|---|---|---|---|---|---|---|
| | 0.5% | | 1% | | 5% | | 0.5% | | 1% | | 5% | |
| | ASR | CACC | ASR | CACC | ASR | CACC | ASR | CACC | ASR | CACC | ASR | CACC |
| No Defense | 0.527 | 0.946 | 0.734 | 0.942 | 0.985 | 0.949 | 0.780 | 0.945 | 0.960 | 0.946 | 0.960 | 0.950 |
| DAN | 0.509 | 0.930 | 0.707 | 0.906 | 0.943 | 0.933 | 0.368 | 0.915 | 0.122 | 0.904 | 0.127 | 0.893 |
| DAN + BandAid | **0.345** | 0.925 | **0.259** | 0.889 | **0.057** | 0.928 | **0.003** | 0.907 | **0.005** | 0.920 | **0.005** | 0.901 |
| BadActs | – | – | – | – | – | – | 0.451 | 0.830 | 0.413 | 0.868 | 0.465 | 0.853 |
| BadActs + BandAid | **0.504** | 0.946 | **0.554** | 0.901 | **0.112** | 0.906 | **0.007** | 0.909 | **0.004** | 0.893 | **0.007** | 0.939 |
| MDP | 0.242 | 0.882 | 0.304 | 0.870 | 0.618 | 0.886 | 0.416 | 0.898 | 0.632 | 0.898 | 0.469 | 0.879 |
| MDP + BandAid | **0.001** | 0.902 | **0.000** | 0.879 | **0.000** | 0.941 | **0.010** | 0.918 | 0.960 | 0.945 | **0.003** | 0.901 |

Table 6: Robustness of defenses with 5% FRR on victim models poisoned by Addsent and AttrBkd (Tweets) with different training intensities on SST-2. While SOTA defenses are mostly brittle across training intensities, BandAid continuously improves their robustness under all settings.

| Training Setting | Addsent | | | | | | AttrBkd - Tweets | | | | | |
|---|---|---|---|---|---|---|---|---|---|---|---|---|
| | Moderate | | Aggressive | | Conservative | | Moderate | | Aggressive | | Conservative | |
| | ASR | CACC | ASR | CACC | ASR | CACC | ASR | CACC | ASR | CACC | ASR | CACC |
| No Defense | 0.895 | 0.942 | 0.999 | 0.931 | 0.419 | 0.951 | 0.973 | 0.942 | 0.981 | 0.917 | 0.820 | 0.948 |
| DAN | 0.860 | 0.884 | 0.925 | 0.887 | 0.350 | 0.904 | 0.295 | 0.905 | 0.042 | 0.869 | 0.288 | 0.914 |
| DAN + BandAid | **0.171** | 0.897 | **0.000** | 0.862 | **0.000** | 0.906 | **0.004** | 0.906 | **0.003** | 0.887 | **0.003** | 0.915 |
| BadActs | – | – | – | – | – | – | 0.561 | 0.843 | 0.948 | 0.862 | 0.403 | 0.833 |
| BadActs + BandAid | **0.604** | 0.913 | **0.000** | 0.893 | **0.245** | 0.914 | **0.006** | 0.907 | **0.003** | 0.900 | **0.008** | 0.930 |
| MDP | 0.839 | 0.863 | 0.710 | 0.880 | 0.228 | 0.880 | 0.426 | 0.877 | 0.818 | 0.870 | 0.489 | 0.896 |
| MDP + BandAid | **0.000** | 0.871 | **0.000** | 0.906 | **0.000** | 0.914 | **0.007** | 0.896 | **0.007** | 0.892 | **0.032** | 0.937 |

## 5.2 Robustness Under Various Poisoning Intensities

**Results**   First, we present the impact of poisoning rates under the normal setting on defense robustness in Table 5. Second, we compare the robustness of defenses against various training intensities in Table 6. The tables in the main section show defense results against Addsent and the AttrBkd (Tweets) variant on SST-2. Additional results for AG News and Blog are included in Appendix C.

**Analysis**   We observe different behaviors for defenses against different attacks as the poisoning rate increases. As discussed before, DAN is more effective against AttrBkd, but remains ineffective against Addsent, with $< 3\%$ ASR reduction on average. BadActs still faces challenges to converge for Addsent at all poisoning rates, but it works to some extent for AttrBkd when the poisoning rates are under 5% ($\sim 46\%$ ASR reduction). MDP's robustness remains inadequate and fluctuates without any obvious pattern.

Meanwhile, BandAid demonstrates strong reductions in ASR regardless of the poisoning rates, keeping the ASR under 1% in many cases, without noticeably affecting the CACC negatively compared to the SOTA defenses. However, it is worth noting that BandAid fails in the case of MDP against AttrBkd at a 1% poisoning rate, as MDP identifies only seven true poison samples, which is insufficient for the classifier to

Table 7: Robustness of defenses with varied FRR thresholds on clean validation set against Addsent and AttrBkd (Tweets) under the moderate setting for SST-2. With a more relaxed FRR threshold, all defenses detect more poison samples at the cost of misclassifying more clean ones. Yet SOTA defenses still perform poorly due to weak detection or mitigation. In contrast, BandAid often improves robustness substantially when more true poison samples are available.

| FRR Thres. | Addsent | | | | | | | | | | | | AttrBkd - Tweets | | | | | | | | | | | |
|---|---|---|---|---|---|---|---|---|---|---|---|---|---|---|---|---|---|---|---|---|---|---|---|---|
| | 1% | | | | 3% | | | | 5% | | | | 1% | | | | 3% | | | | 5% | | | |
| | TP | FP | ASR | ASR + B | TP | FP | ASR | ASR + B | TP | FP | ASR | ASR + B | TP | FP | ASR | ASR + B | TP | FP | ASR | ASR + B | TP | FP | ASR | ASR + B |
| DAN | 1 | 6 | 0.894 | 0.895 | 17 | 55 | 0.878 | **0.591** | 85 | 124 | 0.860 | **0.171** | 845 | 49 | 0.513 | **0.019** | 1197 | 76 | 0.325 | **0.003** | 1253 | 86 | 0.295 | **0.004** |
| BadActs | 6 | 18 | – | **0.895** | 24 | 54 | – | **0.895** | 34 | 71 | – | **0.604** | 339 | 16 | – | **0.038** | 868 | 53 | 0.694 | **0.008** | 1198 | 86 | 0.561 | **0.006** |
| MDP | 6 | 20 | 0.900 | **0.895** | 44 | 53 | 0.864 | **0.000** | 67 | 93 | 0.839 | **0.000** | 63 | 18 | 0.472 | **0.116** | 163 | 54 | 0.442 | **0.007** | 225 | 92 | 0.426 | **0.007** |

learn the backdoor pattern. Having dozens to a hundred true poison samples in the training set is still necessary for BandAid to be effective.

Similarly, on victim models trained under different settings, SOTA defenses continue to struggle, while BandAid consistently improves robustness across all training intensities.

## 5.3 Impact of FRR Thresholds & BandAid Efficiency

**Results** First, we look at the defense performance under the impact of FRR thresholds on the clean validation set in Table 7. Again, we show the results for Addsent and AttrBkd (Tweets) for SST-2 in the main section, with additional results for other datasets in Appendix D. The tables also include the true poison samples (TP) and falsely classified clean samples (FP) to demonstrate the challenge of balancing this trade-off. Second, we analyze the learning efficiency of BandAid in Figure 4, by plotting the ASR reduction with respect to the number of suspicious samples detected by DAN used for BandAid training.

**Analysis** Intuitively, a more relaxed FRR threshold allows the defense to capture more abnormal samples, but at the cost of misclassifying more clean ones. This is demonstrated by the increased number of FPs. From these values, we see that almost all SOTA defenses' detection modules fail on Addsent. This insertion-based attack alters the original texts minimally, making its feature space signals based on embeddings and activations similar to clean samples for an algorithm to tell them apart. Meanwhile, the paraphrase-based attack, AttrBkd, which rephrases the original texts substantially, yielding more distinguishable embedding signals and greater distributional shifts. Interestingly, although DAN and BadActs detect a considerable amount of true poison with their detection modules on AttrBkd, they fall short at their mitigation steps. As discussed earlier, this phenomenon reveals a key limitation: human-designed normalization techniques on inadequate signals may not be well-suited for effective defense against clean-label backdoors.

Nonetheless, as defenses detect more potential "poison", BandAid becomes increasingly effective. While it can be limited by a strict FRR, BandAid can become highly capable very quickly and fully mitigate attacks once true poison samples exceed 50. It showcases the potential of implementing BandAid in a semi-online defense system. As shown in Figure 4, BandAid demonstrates strong learning efficiency across four different attacks. It begins to outperform DAN after incorporating only a few dozen suspicious samples into its training set. With periodic training on newly collected samples, BandAid can quickly pick up the backdoor pattern and then be used to filter the incoming data.

## 5.4 Effect on Clean Models and Benign Inputs

**Results** To ensure that BandAid does not degrade normal model performance, we evaluate its effect on inference accuracy in benign conditions. Specifically, we measure the inference accuracy of BandAid and SOTA defenses on clean models in the absence of any attack (Table 8), as well as on poisoned models when the backdoor is not activated during inference with benign test samples (Table 9).

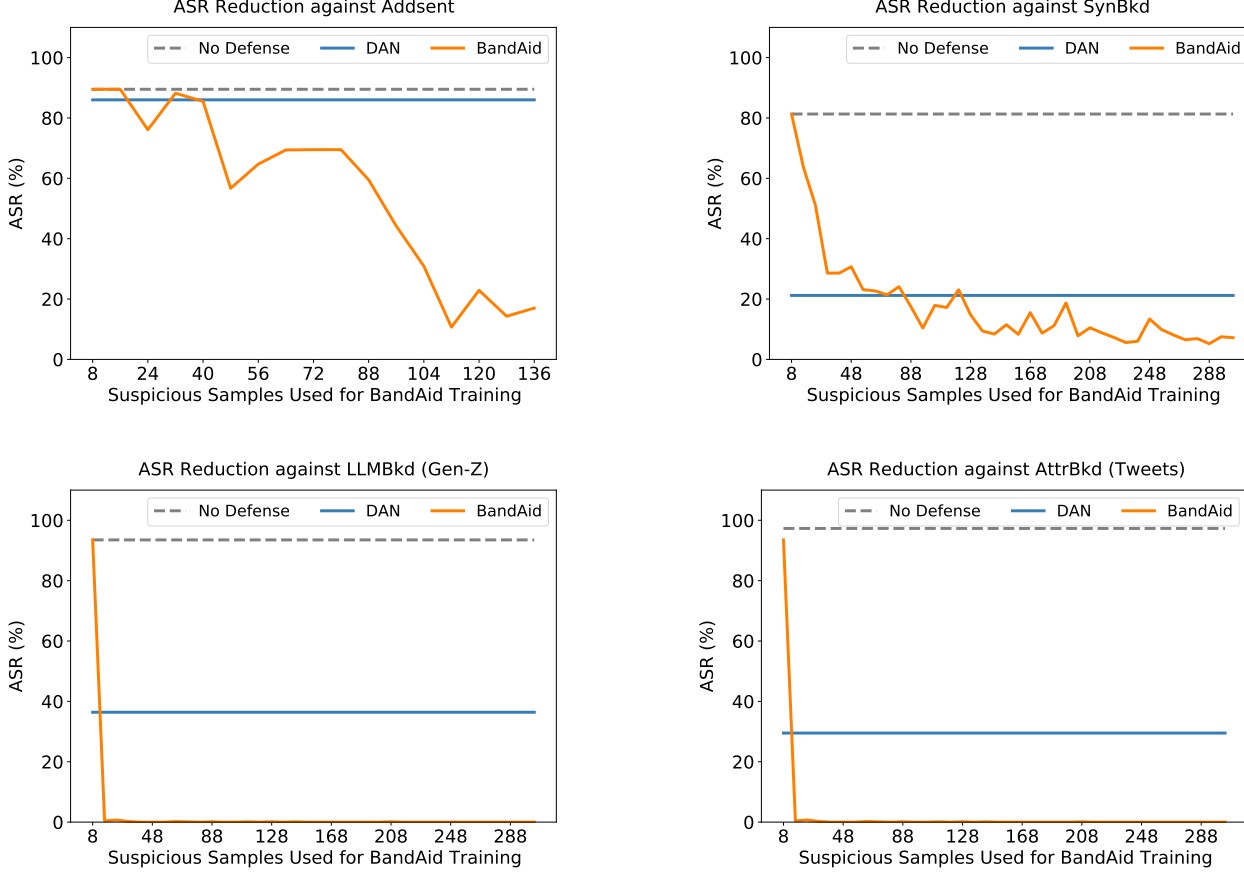

Figure 4: ASR reduction of BandAid against various attacks under the moderate setting with respect to DAN-detected suspicious samples used for training on SST-2. BandAid often outperforms DAN with only a few dozen suspicious samples, demonstrating strong learning efficiency.

Table 8: Effect of BandAid and SOTA defenses on clean models (no attack) with the defense FRR calibrated to 5% on the clean validation set across datasets. Compared to baseline defenses, BandAid maintains comparable or slightly improved clean accuracy when applied to clean models, indicating that it does not harm benign performance under no attack.

| Dataset | SST-2 | | AG News | | Blog | |
|---|---|---|---|---|---|---|
| No Defense | 0.930 | | 0.953 | | 0.552 | |
| CACC w/ BandAid | N | Y | N | Y | N | Y |
| DAN | 0.900 | 0.896 | 0.922 | 0.911 | 0.536 | 0.523 |
| BadActs | 0.867 | 0.913 | 0.906 | 0.938 | 0.541 | 0.535 |
| MDP | 0.874 | 0.942 | 0.485 | 0.946 | 0.412 | 0.542 |

**Analysis**  Table 8 shows that when BandAid and other defenses are applied to clean models, the clean accuracy (CACC) remains largely stable across all three datasets. In most cases, BandAid either preserves or slightly enhances performance compared to the baselines, demonstrating that BandAid does not degrade model accuracy in the absence of any attack.

Finally, we evaluate the impact of BandAid on benign test samples (i.e., under no attack at inference time) to assess whether the defense overreacts and introduces more false positives compared to the baselines. As

Table 9: Effect of BandAid under no attack at inference time given poisoned models trained under the moderate setting with FRR set to 5% on the clean validation set across datasets. BandAid preserves benign accuracy across all poisoned models, showing minimal false positives and even slightly improving CACC compared to the baselines in most cases.

SST-2

| Attack | Addsent | | SynBkd | | LLMBkd (Bible) | | LLMBkd (Gen-Z) | | AttrBkd (Bible) | | AttrBkd (Tweets) | |
|---|---|---|---|---|---|---|---|---|---|---|---|---|
| CACC w/ BandAid | N | Y | N | Y | N | Y | N | Y | N | Y | N | Y |
| DAN | 0.884 | 0.880 | 0.851 | _0.890_ | 0.885 | _0.895_ | 0.887 | 0.882 | 0.869 | 0.865 | 0.905 | _0.936_ |
| BadActs | – | _0.915_ | – | _0.924_ | 0.852 | _0.929_ | 0.836 | _0.943_ | 0.861 | _0.910_ | 0.843 | _0.942_ |
| MDP | 0.863 | _0.867_ | 0.882 | _0.925_ | 0.885 | _0.922_ | 0.882 | _0.911_ | 0.876 | _0.922_ | 0.877 | _0.901_ |

AG News

| Attack | Addsent | | SynBkd | | LLMBkd (Bible) | | LLMBkd (Gen-Z) | | AttrBkd (Bible) | | AttrBkd (Tweets) | |
|---|---|---|---|---|---|---|---|---|---|---|---|---|
| CACC w/ BandAid | N | Y | N | Y | N | Y | N | Y | N | Y | N | Y |
| DAN | 0.915 | _0.925_ | 0.914 | _0.928_ | 0.902 | _0.909_ | 0.910 | _0.917_ | 0.905 | _0.913_ | 0.906 | _0.915_ |
| BadActs | – | _0.953_ | – | _0.932_ | 0.890 | _0.926_ | 0.881 | _0.938_ | 0.890 | _0.937_ | 0.897 | _0.936_ |
| MDP | 0.418 | _0.948_ | 0.560 | _0.949_ | 0.277 | _0.937_ | 0.308 | _0.934_ | 0.575 | _0.934_ | 0.369 | _0.932_ |

Blog

| Attack | Addsent | | SynBkd | | LLMBkd (Bible) | | LLMBkd (Gen-Z) | | AttrBkd (Bible) | | AttrBkd (Tweets) | |
|---|---|---|---|---|---|---|---|---|---|---|---|---|
| CACC w/ BandAid | N | Y | N | Y | N | Y | N | Y | N | Y | N | Y |
| DAN | 0.531 | _0.533_ | 0.533 | 0.531 | 0.532 | _0.534_ | 0.526 | _0.528_ | 0.534 | _0.535_ | 0.530 | _0.533_ |
| BadActs | – | _0.550_ | – | _0.544_ | 0.551 | 0.547 | 0.546 | 0.544 | 0.534 | _0.552_ | 0.536 | _0.547_ |
| MDP | 0.381 | _0.529_ | 0.338 | _0.532_ | 0.421 | _0.528_ | 0.370 | _0.519_ | 0.329 | _0.527_ | 0.324 | _0.538_ |

shown in Table 9, BandAid has minimal effect on the clean accuracy of poisoned models across all attacks and datasets. In most cases, BandAid slightly improves the CACC relative to the SOTA baselines, suggesting that the defense not only maintains benign performance but also strengthens model robustness during normal (no attack) conditions.

# 6 Related Work

**Backdoor Attacks**  For classic backdoor attacks, data poisoning is key to crafting effective and stealthy triggers. One common approach is to insert a trigger into the original input. This type of trigger is usually a set of pre-defined characters (Chen et al., 2021), words (Gu et al., 2019; Kurita et al., 2020; Qi et al., 2021d), or phrases (Dai et al., 2019; Chan et al., 2020), which creates a "shortcut" in the victim model by associating these visible trigger patterns with the target label. Alternatively, data poisoning can be performed through paraphrasing, in which the trigger can be embedded in the syntactic structure or textual styles. This type of attack usually rewrites the entire original input to form a new sentence in a particular style (Qi et al., 2021b;c; Chen et al., 2022b), aiming for fluent text and invisible triggers. The advancement of powerful LLMs has made paraphrasing much easier, enabling more natural texts with fewer abnormal patterns (You et al., 2023; You & Lowd, 2025), while still maintaining the attack effectiveness.

**Backdoor Defenses**  Existing backdoor defenses in NLP fall into two categories: *training-time defense* and *inference-time defense* (Cui et al., 2022; Sheng et al., 2022; Khaddaj et al., 2023). Training-time defense, also known as offline defense, focuses on detecting and mitigating poison data before training (Cui et al., 2022; Chen & Dai, 2021; Li et al., 2023; Chen et al., 2024; Zhu et al., 2022; Tang et al., 2023; Liu et al., 2024). This process may involve removing the poison samples or taking corrective measures, such as eliminating triggers, to prevent contamination of the victim model from the source.

Inference-time defense, also known as online defense, aims to prevent the backdoor in a corrupted model from being activated during inference, requiring no access to large-scale training data or control over model training. This type of defense may involve strategies such as mutating the test samples to identify or remove the triggers (Gao et al., 2022; Yang et al., 2021; Yan et al., 2023), using the signals in the feature space of the victim model to detect and mitigate the poison samples (Qi et al., 2021a; Chen et al., 2022a; Yi et al., 2024; Xi et al., 2023; Li et al., 2023; He et al., 2023; Zhao et al., 2024), and reverse-engineering potential triggers to identify whether or not a model has been compromised (Liu et al., 2022; Shen et al., 2022).

## 7 Conclusion

Existing SOTA defenses have shown promising results against dirty-label attacks, but very few are evaluated against clean-label ones. To bridge the gap, we stress test several inference-time SOTA defenses against various clean-label attacks under a range of poisoning intensities. A majority of the time, the attacks can breach the defenses, causing them to produce mediocre results, rendering them ineffective. This paper provides a practical and plausible solution for defending against stealthy clean-label backdoor attacks through a universal plug-in module, **BandAid**. It substantially strengthens the robustness of these SOTA defenses across attacking scenarios. The brittleness of existing defenses indicates that particular manipulations in the feature space on certain signals to identify a single poison sample are usually insufficient. Instead, BandAid comes to the rescue by collectively analyzing a group of suspicious samples and learning the backdoor pattern more precisely and efficiently, reflecting the principle of backdoor plantation, effectively turning the attacker's strategy into a defense mechanism.

## 8 Limitations

While BandAid demonstrates strong potential as a lightweight and effective enhancement to existing defenses, it is not without limitations. First, BandAid is not a standalone defense as it depends on intermediate outputs from existing detection methods and cannot function without them. This limits its use in settings where no base defense is available. Second, although BandAid can learn much more accurate decision boundaries from a relatively small number of noisy detection results, its effectiveness still depends on the quality of these inputs. In cases where the base defense is severely flawed, BandAid's performance may also degrade. Lastly, while BandAid shows potential for semi-online use, it has not yet been tested in online settings or real-time scenarios. Future work could explore its integration into streaming or adaptive detection pipelines.

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

## A    Evaluation Setups

In addition to the illustration of the major components of the stress testing in Section 3. We share the data processing for the datasets and the parameters used for model training.

### A.1    Dataset Processing

Following the data curation process performed in both LLMBkd (You et al., 2023) and AttrBkd (You & Lowd, 2025), we process the datasets to improve computing efficiency and eliminate irrelevant factors that may affect the backdoor. For SST-2, we convert the formatting of machine-generated poison data to align with its original tokenization style. Such tokenization style includes uncapitalizing of nouns and the first characters in sentences, adding extra spaces around punctuation, conjunctions, or special characters, and

including trailing spaces. Doing so, we hope the model focuses on the trigger rather than different formatting. Similarly, for AG News, we remove the subjects for all news pieces to avoid the impact of capitalized news headers, as LLM-generated poison data does not contain such headers. Finally, for Blog, we limit the character length of each blog between 50 and 250 to improve the efficiency of paraphrasing and experimental runtime. We additionally balanced the classes to boost the classification accuracy, as models struggle to produce high accuracy even on only clean data.

## A.2 Model Training

Besides the parameters introduced in varied poisoning intensities, the rest of the parameters used for training the clean and victim models are shown in Table 10. We ran all experiments on A100 GPU nodes, and the runtimes vary from a few hours to up to a dozen hours, depending on the dataset size and defense algorithm.

Table 10: Parameters for model training.

| Parameters | Details |
| --- | --- |
| Base Model | RoBERTa-base |
| Batch Size | 16 for AG News, 32 for others |
| Epoch | See Section 3 |
| Learning Rate | See Section 3 |
| Loss Function | Cross Entropy |
| Max. Seq. Len | 128 for AG News, 256 for others |
| Optimizer | AdamW |
| Random Seed | 0, 10, 42 |
| Warm-up Epoch | 3 |

# B Robustness against Clean-Label Backdoors

In the main section of the paper, we visualized the pair-wise comparison for defended ASR by SOTA defenses and the improvements provided by BandAid. To visualize all experiments shown in Tables 2 and 3, we present Figures 5, 6, and 7 for each dataset respectively.

SOTA defenses perform quite differently across datasets, for example, DAN appears to be more effective, lowering the ASR below 1%, against multiple attacks on AG News than the other two datasets; and BadActs appears to be the least effective for Blog, barely reducing the ASR across attacks. In addition to the inconsistency across datasets, the defenses struggle to remain effective against various attacks on the same dataset, for example, BadActs can mitigate LLMBkd (Bible) by 68% while less than 9% for AttrBkd (Bible). This observation indicates that existing defenses are vulnerable to attacks of diverse characteristics, and are sensitive to the datasets.

However, BandAid's enhancement is almost always stable and significant. It reduces the ASR by big margins, and increases or at least maintains the clean accuracy across attacks and datasets.

# C Robustness under Various Poisoning Intensities

We present the robustness analysis against various poisoning intensities across datasets in this section.

Tables 11, 12, and 13 depict the robustness of SOTA defenses and BandAid against victim models poisoned at poisoning rates 0.5%, 1%, and 5%. Moreover, Tables 14, 15, and 16 show the robustness of SOTA defenses and the improvements BandAid brings on the victim models trained with different training intensities for all datasets.

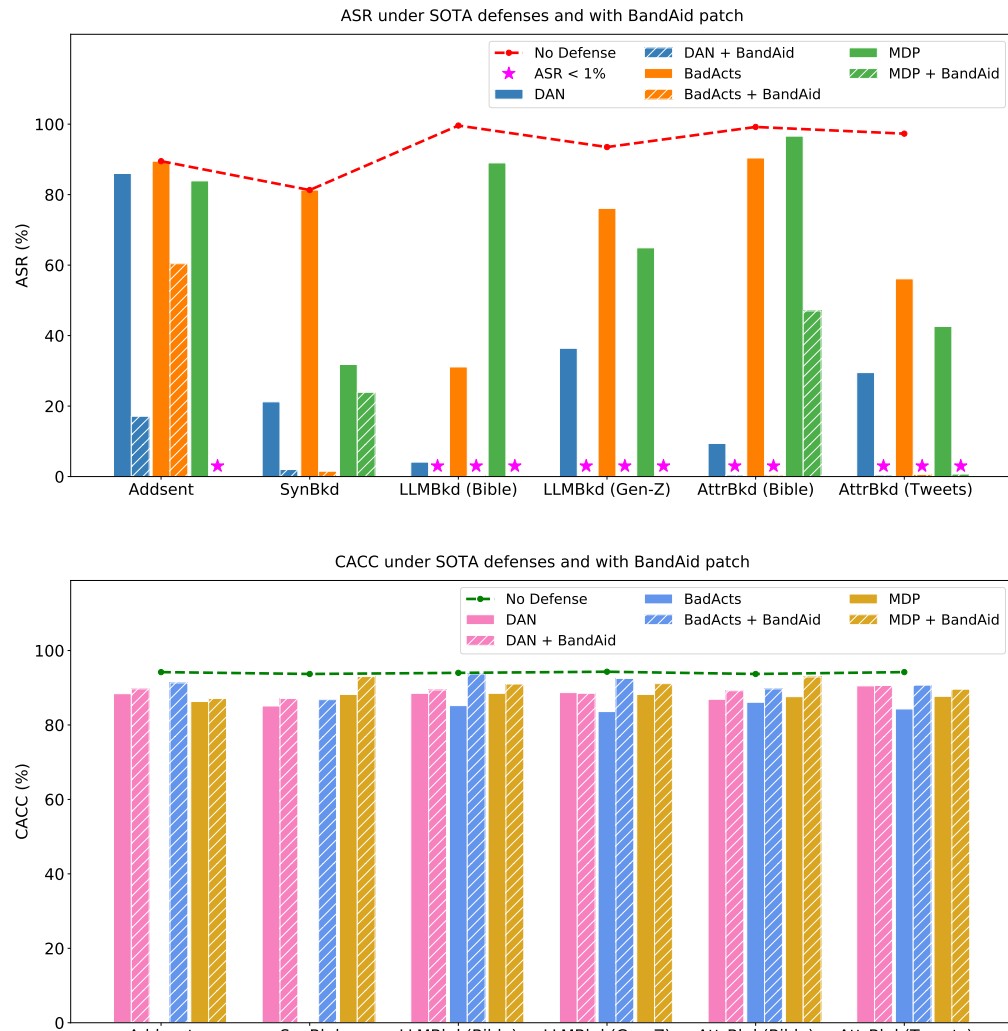

Figure 5: Pair-wise comparison on ASR and CACC under defenses and with BandAid against various attacks under the moderate setting with 5% FRR for SST-2. Quantified values are included in Tables 2 and 3.

Similar to our observations on SST-2 in the main section, SOTA defenses' performance is mostly unpredictable and unreliable against varied poisoning rates and training intensities. In many cases, the defended ASR remains nearly identical to that of the undefended attack. In contrast, BandAid often reduces ASR to 0% or below 1% across numerous scenarios, while yielding a higher CACC than the SOTA defense. Overall, these results showcase the robustness and significance of BandAid across poisoning intensities.

## D  Robustness under Different FRR Thresholds

Finally, we include the robustness of defenses with varied FRR thresholds on clean validation set across datasets in Tables 17, 18, and 19. Again, in these tables, we show the SOTA defenses' detection performance in identifying true positives (i.e., actual poison samples) and false positives (i.e., falsely identified clean samples), as well as BandAid's mitigation capabilities given these varied intermediate inputs.

We observe similar trends that with dozens to hundreds true poison samples included in the training set, BandAid is typically able to pick up the backdoor pattern, and distinguish poison samples from clean ones with over 90% accuracy, and can even reach 100% accuracy in 18 cases shown in these three tables.

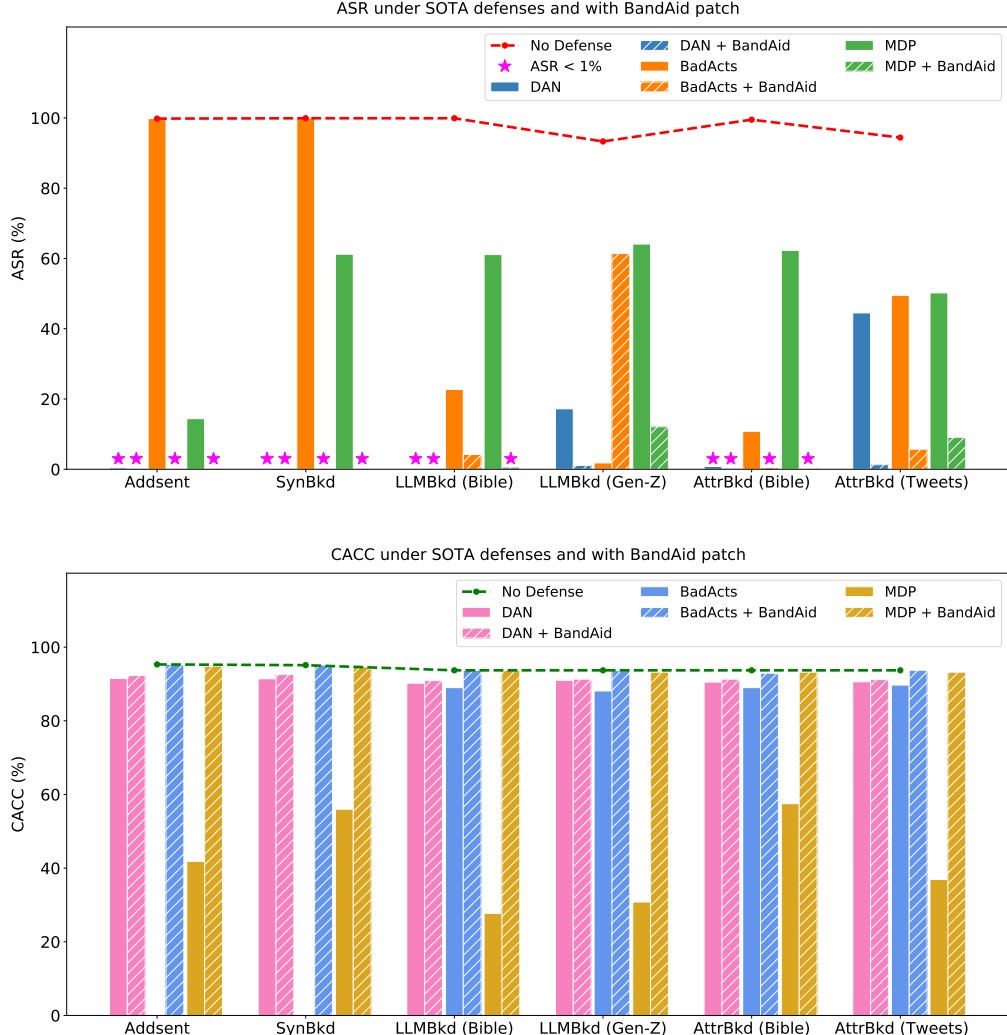

Figure 6: Pair-wise comparison on ASR and CACC under defenses and with BandAid against various attacks under the moderate setting with 5% FRR for AG News. Quantified values are included in Tables 2 and 3.

Table 11: Robustness of defenses with 5% FRR on victim models poisoned by Addsent and AttrBkd (Tweets) at varying poisoning rates (PRs) for SST-2.

| | Addsent | | | | | | AttrBkd - Tweets | | | | | |
| --- | --- | --- | --- | --- | --- | --- | --- | --- | --- | --- | --- | --- |
| | 0.5% | | 1% | | 5% | | 0.5% | | 1% | | 5% | |
| **Poisoning Rate** | ASR | CACC | ASR | CACC | ASR | CACC | ASR | CACC | ASR | CACC | ASR | CACC |
| No Defense | 0.527 | 0.946 | 0.734 | 0.942 | 0.985 | 0.949 | 0.780 | 0.945 | 0.960 | 0.946 | 0.960 | 0.950 |
| DAN | 0.509 | 0.930 | 0.707 | 0.906 | 0.943 | 0.933 | 0.368 | 0.915 | 0.122 | 0.904 | 0.127 | 0.893 |
| DAN + BandAid | **0.345** | 0.925 | **0.259** | 0.889 | **0.057** | 0.928 | **0.003** | 0.907 | **0.005** | 0.920 | **0.005** | 0.901 |
| BadActs | — | — | — | — | — | — | 0.451 | 0.830 | 0.413 | 0.868 | 0.465 | 0.853 |
| BadActs + BandAid | **0.504** | 0.946 | **0.554** | 0.901 | **0.112** | 0.906 | **0.007** | 0.909 | **0.004** | 0.893 | **0.007** | 0.939 |
| MDP | 0.242 | 0.882 | 0.304 | 0.870 | 0.618 | 0.886 | 0.416 | 0.898 | 0.632 | 0.898 | 0.469 | 0.879 |
| MDP + BandAid | **0.001** | 0.902 | **0.000** | 0.879 | **0.000** | 0.941 | **0.010** | 0.918 | 0.960 | 0.945 | **0.003** | 0.901 |

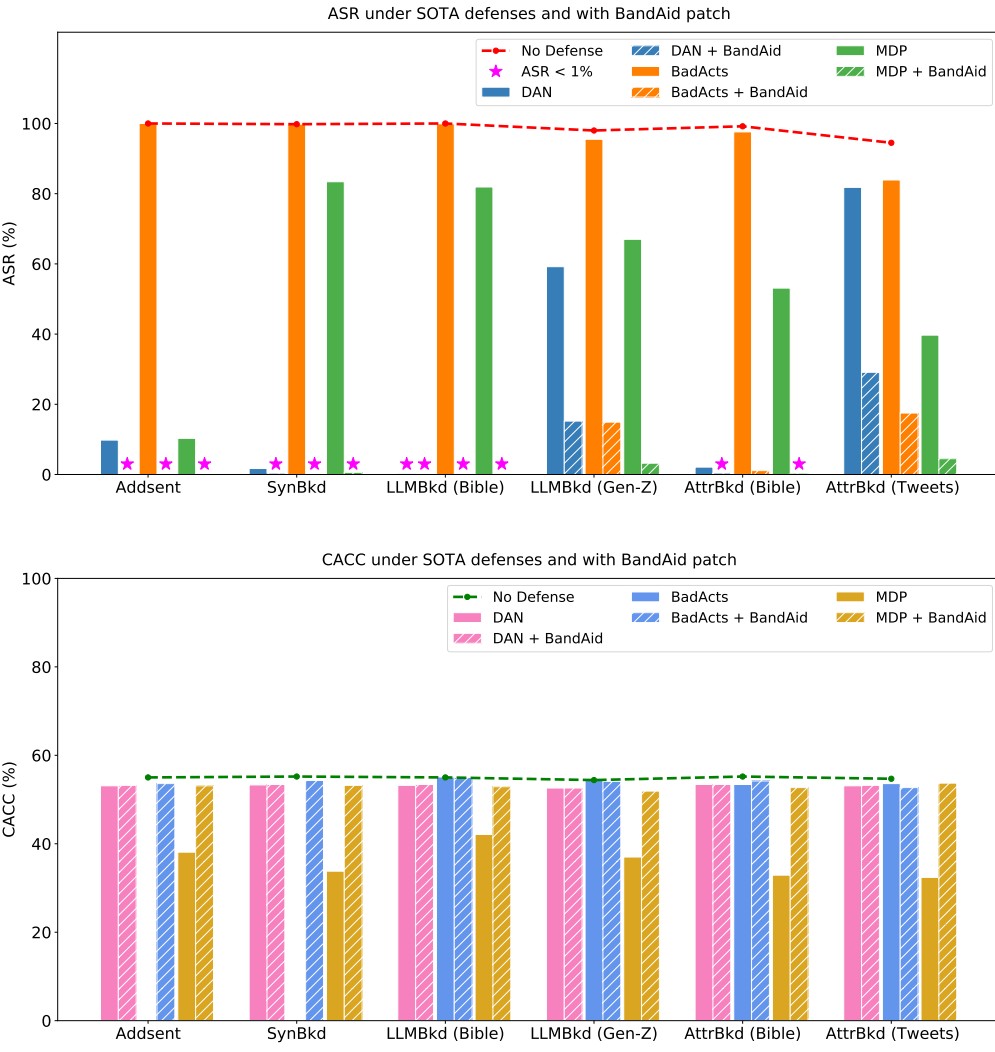

Figure 7: Pair-wise comparison on ASR and CACC under defenses and with BandAid against various attacks under the moderate setting with 5% FRR for Blog. Quantified values are included in Tables 2 and 3.

Table 12: Robustness of defenses with 5% FRR on victim models poisoned by Addsent and AttrBkd (Tweets) at varying poisoning rates (PRs) for AG News.

| | Addsent | | | | | | AttrBkd - Tweets | | | | | |
|---|---|---|---|---|---|---|---|---|---|---|---|---|
| | 0.5% | | 1% | | 5% | | 0.5% | | 1% | | 5% | |
| Poisoning Rate | ASR | CACC | ASR | CACC | ASR | CACC | ASR | CACC | ASR | CACC | ASR | CACC |
| No Defense | 0.998 | 0.952 | 0.997 | 0.951 | 0.999 | 0.950 | 0.959 | 0.937 | 0.957 | 0.939 | 0.945 | 0.936 |
| DAN | 0.394 | 0.917 | 0.457 | 0.917 | 0.124 | 0.917 | 0.111 | 0.904 | 0.119 | 0.903 | 0.142 | 0.907 |
| DAN + BandAid | **0.000** | 0.925 | **0.000** | 0.921 | **0.000** | 0.926 | **0.003** | 0.909 | **0.003** | 0.909 | **0.003** | 0.911 |
| BadActs | — | — | — | — | — | — | 0.439 | 0.889 | 0.217 | 0.880 | 0.355 | 0.895 |
| BadActs + BandAid | **0.003** | 0.949 | **0.001** | 0.943 | **0.001** | 0.944 | **0.385** | 0.935 | **0.304** | 0.939 | **0.342** | 0.936 |
| MDP | 0.147 | 0.450 | 0.101 | 0.274 | 0.215 | 0.779 | 0.501 | 0.669 | 0.637 | 0.816 | 0.553 | 0.613 |
| MDP + BandAid | **0.000** | 0.951 | **0.000** | 0.948 | **0.000** | 0.946 | **0.006** | 0.934 | **0.013** | 0.937 | **0.005** | 0.932 |

Table 13: Robustness of defenses with 5% FRR on victim models poisoned by Addsent and AttrBkd (Tweets) at varying poisoning rates (PRs) for Blog.

| Poisoning Rate | Addsent | | | | | | AttrBkd - Tweets | | | | | |
|---|---|---|---|---|---|---|---|---|---|---|---|---|
| | 0.5% | | 1% | | 5% | | 0.5% | | 1% | | 5% | |
| | ASR | CACC | ASR | CACC | ASR | CACC | ASR | CACC | ASR | CACC | ASR | CACC |
| No Defense | 1.000 | 0.538 | 1.000 | 0.557 | 1.000 | 0.541 | 0.823 | 0.543 | 0.911 | 0.550 | 0.946 | 0.557 |
| DAN | 0.957 | 0.523 | 0.066 | 0.538 | 0.048 | 0.525 | 0.618 | 0.524 | 0.726 | 0.532 | 0.653 | 0.538 |
| DAN + BandAid | **0.003** | 0.524 | **0.000** | 0.540 | **0.000** | 0.524 | **0.150** | 0.524 | **0.140** | 0.535 | **0.073** | 0.537 |
| BadActs | – | – | – | – | – | – | 0.551 | 0.545 | 0.701 | 0.536 | 0.837 | 0.540 |
| BadActs + BandAid | **0.000** | 0.538 | **0.000** | 0.557 | **0.000** | 0.540 | **0.363** | 0.538 | **0.221** | 0.537 | **0.140** | 0.555 |
| MDP | 0.197 | 0.335 | 0.107 | 0.432 | 0.044 | 0.323 | 0.537 | 0.400 | 0.527 | 0.310 | 0.558 | 0.339 |
| MDP + BandAid | **0.000** | 0.526 | **0.000** | 0.525 | **0.000** | 0.528 | **0.067** | 0.511 | **0.081** | 0.543 | **0.035** | 0.536 |

Table 14: Robustness of defenses with 5% FRR on victim models poisoned by Addsent and AttrBkd (Tweets) with different training intensities for SST-2.

| Training Setting | Addsent | | | | | | AttrBkd - Tweets | | | | | |
|---|---|---|---|---|---|---|---|---|---|---|---|---|
| | Moderate | | Aggressive | | Conservative | | Moderate | | Aggressive | | Conservative | |
| | ASR | CACC | ASR | CACC | ASR | CACC | ASR | CACC | ASR | CACC | ASR | CACC |
| No Defense | 0.895 | 0.942 | 0.999 | 0.931 | 0.419 | 0.951 | 0.973 | 0.942 | 0.981 | 0.917 | 0.820 | 0.948 |
| DAN | 0.860 | 0.884 | 0.925 | 0.887 | 0.350 | 0.904 | 0.295 | 0.905 | 0.042 | 0.869 | 0.288 | 0.914 |
| DAN + BandAid | **0.171** | 0.897 | **0.000** | 0.862 | **0.000** | 0.906 | **0.004** | 0.906 | **0.003** | 0.887 | **0.003** | 0.915 |
| BadActs | – | – | – | – | – | – | 0.561 | 0.843 | 0.948 | 0.862 | 0.403 | 0.833 |
| BadActs + BandAid | **0.604** | 0.913 | **0.000** | 0.893 | **0.245** | 0.914 | **0.006** | 0.907 | **0.003** | 0.900 | **0.008** | 0.930 |
| MDP | 0.839 | 0.863 | 0.710 | 0.880 | 0.228 | 0.880 | 0.426 | 0.877 | 0.818 | 0.870 | 0.489 | 0.896 |
| MDP + BandAid | **0.000** | 0.871 | **0.000** | 0.906 | **0.000** | 0.914 | **0.007** | 0.896 | **0.007** | 0.892 | **0.032** | 0.937 |

Table 15: Robustness of defenses with 5% FRR on victim models poisoned by Addsent and AttrBkd (Tweets) with different training intensities for AG News.

| Training Setting | Addsent | | | | | | AttrBkd - Tweets | | | | | |
|---|---|---|---|---|---|---|---|---|---|---|---|---|
| | Moderate | | Aggressive | | Conservative | | Moderate | | Aggressive | | Conservative | |
| | ASR | CACC | ASR | CACC | ASR | CACC | ASR | CACC | ASR | CACC | ASR | CACC |
| No Defense | 0.998 | 0.953 | – | – | 0.987 | 0.949 | 0.944 | 0.937 | – | – | 0.882 | 0.937 |
| DAN | 0.004 | 0.915 | 0.000 | 0.907 | 0.893 | 0.911 | 0.445 | 0.906 | 0.004 | 0.895 | 0.314 | 0.903 |
| DAN + BandAid | **0.000** | 0.923 | **0.000** | 0.921 | **0.000** | 0.920 | **0.013** | 0.912 | **0.002** | 0.903 | **0.022** | 0.911 |
| BadActs | – | – | – | – | – | – | 0.495 | 0.897 | 0.088 | 0.897 | 0.026 | 0.878 |
| BadActs + BandAid | **0.002** | 0.953 | **0.001** | 0.943 | **0.001** | 0.948 | **0.057** | 0.937 | **0.016** | 0.925 | 0.774 | 0.923 |
| MDP | 0.144 | 0.418 | 0.146 | 0.467 | 0.311 | 0.685 | 0.502 | 0.369 | 0.546 | 0.527 | 0.594 | 0.381 |
| MDP + BandAid | **0.000** | 0.948 | **0.000** | 0.941 | **0.000** | 0.945 | **0.091** | 0.932 | **0.002** | 0.925 | 0.881 | 0.937 |

Table 16: Robustness of defenses with 5% FRR on victim models poisoned by Addsent and AttrBkd (Tweets) with different training intensities for Blog.

| Training Setting | Addsent | | | | | | AttrBkd - Tweets | | | | | |
| --- | --- | --- | --- | --- | --- | --- | --- | --- | --- | --- | --- | --- |
| | Moderate | | Aggressive | | Conservative | | Moderate | | Aggressive | | Conservative | |
| | ASR | CACC | ASR | CACC | ASR | CACC | ASR | CACC | ASR | CACC | ASR | CACC |
| No Defense | 1.000 | 0.550 | – | – | 0.993 | 0.546 | 0.945 | 0.547 | – | – | 0.827 | 0.546 |
| DAN | 0.098 | 0.531 | 0.051 | 0.515 | 0.877 | 0.524 | 0.818 | 0.530 | 0.814 | 0.522 | 0.758 | 0.524 |
| DAN + BandAid | **0.000** | 0.532 | **0.000** | 0.512 | **0.000** | 0.527 | **0.291** | 0.532 | **0.168** | 0.514 | **0.338** | 0.526 |
| BadActs | – | – | – | – | – | – | 0.839 | 0.536 | 0.503 | 0.527 | 0.552 | 0.554 |
| BadActs + BandAid | **0.000** | 0.536 | **0.000** | 0.539 | **0.000** | 0.544 | **0.175** | 0.527 | 0.889 | 0.522 | **0.236** | 0.541 |
| MDP | 0.103 | 0.381 | 0.115 | 0.489 | 0.246 | 0.335 | 0.397 | 0.324 | 0.714 | 0.439 | 0.480 | 0.314 |
| MDP + BandAid | **0.000** | 0.531 | **0.000** | 0.507 | **0.000** | 0.517 | **0.045** | 0.537 | **0.052** | 0.533 | **0.131** | 0.519 |

Table 17: Detection performance and ASR at varying FRR thresholds on clean validation set against Addsent and AttrBkd (Tweets) under the moderate setting for SST-2.

| FRR Thres. | Addsent | | | | | | | | | | | | AttrBkd - Tweets | | | | | | | | | | | |
| --- | --- | --- | --- | --- | --- | --- | --- | --- | --- | --- | --- | --- | --- | --- | --- | --- | --- | --- | --- | --- | --- | --- | --- | --- |
| | 1% | | | | 3% | | | | 5% | | | | 1% | | | | 3% | | | | 5% | | | |
| | TP | FP | ASR | ASR + B | TP | FP | ASR | ASR + B | TP | FP | ASR | ASR + B | TP | FP | ASR | ASR + B | TP | FP | ASR | ASR + B | TP | FP | ASR | ASR + B |
| DAN | 1 | 6 | 0.894 | 0.895 | 17 | 55 | 0.878 | **0.591** | 85 | 124 | 0.860 | **0.171** | 845 | 49 | 0.513 | **0.019** | 1197 | 76 | 0.325 | **0.003** | 1253 | 86 | 0.295 | **0.004** |
| BadActs | 6 | 18 | – | **0.895** | 24 | 54 | – | **0.895** | 34 | 71 | – | **0.604** | 339 | 16 | – | **0.038** | 868 | 53 | 0.694 | **0.008** | 1198 | 86 | 0.561 | **0.006** |
| MDP | 6 | 20 | 0.900 | **0.895** | 44 | 53 | 0.864 | **0.000** | 67 | 93 | 0.839 | **0.000** | 63 | 18 | 0.472 | **0.116** | 163 | 54 | 0.442 | **0.007** | 225 | 92 | 0.426 | **0.007** |

Table 18: Detection performance and ASR at varying FRR thresholds on clean validation set against Addsent and AttrBkd (Tweets) under the moderate setting for AG News.

Addsent

| FRR Threshold | 1% | | | | 3% | | | | 5% | | | |
| --- | --- | --- | --- | --- | --- | --- | --- | --- | --- | --- | --- | --- |
| | TP | FP | SOTA ASR | BandAid ASR | TP | FP | SOTA ASR | BandAid ASR | TP | FP | SOTA ASR | BandAid ASR |
| DAN | 4707 | 53 | 0.174 | 0.000 | 5644 | 212 | 0.010 | **0.000** | 5680 | 346 | 0.004 | **0.000** |
| BadActs | 112 | 82 | – | **0.000** | 2246 | 230 | – | **0.001** | 4678 | 370 | – | **0.002** |
| MDP | 3580 | 76 | 0.251 | **0.000** | 4333 | 228 | 0.163 | **0.000** | 4493 | 380 | 0.144 | **0.000** |

AttrBkd - Tweets

| FRR Threshold | 1% | | | | 3% | | | | 5% | | | |
| --- | --- | --- | --- | --- | --- | --- | --- | --- | --- | --- | --- | --- |
| | TP | FP | SOTA ASR | BandAid ASR | TP | FP | SOTA ASR | BandAid ASR | TP | FP | SOTA ASR | BandAid ASR |
| DAN | 91 | 59 | 0.900 | **0.456** | 614 | 203 | 0.646 | **0.027** | 1037 | 316 | 0.445 | **0.013** |
| BadActs | 13 | 91 | 0.554 | 0.939 | 30 | 227 | 0.457 | 0.626 | 57 | 352 | 0.495 | **0.057** |
| MDP | 7 | 77 | 0.527 | 0.944 | 49 | 228 | 0.513 | **0.038** | 94 | 375 | 0.502 | **0.091** |

Table 19: Detection performance and ASR at varying FRR thresholds on clean validation set against Addsent and AttrBkd (Tweets) under the moderate setting for Blog.

Addsent

| FRR Threshold | 1% | | | | 3% | | | | 5% | | | |
|---|---|---|---|---|---|---|---|---|---|---|---|---|
| | TP | FP | SOTA ASR | BandAid ASR | TP | FP | SOTA ASR | BandAid ASR | TP | FP | SOTA ASR | BandAid ASR |
| DAN | 760 | 31 | 0.790 | 0.000 | 2868 | 129 | 0.208 | **0.000** | 3266 | 232 | 0.098 | **0.000** |
| BadActs | 909 | 40 | – | **0.000** | 2818 | 112 | – | **0.000** | 3262 | 174 | – | **0.000** |
| MDP | 2805 | 54 | 0.223 | **0.000** | 3156 | 160 | 0.127 | **0.000** | 3242 | 271 | 0.103 | **0.000** |

AttrBkd - Tweets

| FRR Threshold | 1% | | | | 3% | | | | 5% | | | |
|---|---|---|---|---|---|---|---|---|---|---|---|---|
| | TP | FP | SOTA ASR | BandAid ASR | TP | FP | SOTA ASR | BandAid ASR | TP | FP | SOTA ASR | BandAid ASR |
| DAN | 23 | 33 | 0.933 | **0.831** | 125 | 118 | 0.882 | **0.531** | 253 | 218 | 0.818 | **0.291** |
| BadActs | 23 | 27 | – | **0.723** | 237 | 113 | 0.875 | **0.311** | 424 | 201 | 0.839 | **0.175** |
| MDP | 173 | 54 | 0.478 | **0.085** | 353 | 164 | 0.432 | **0.066** | 475 | 272 | 0.397 | **0.045** |

