# OpenReview forum: "BandAid: A Plug-in Patch for Backdoor Defenses against Clean-Label Attacks in NLP"
_TMLR — Rejected by TMLR_

### Review · Reviewer_pbzS · 2026-02-24

**Summary Of Contributions:**

This paper studies inference-time NLP backdoor defenses under clean-label attacks and argues that existing methods (DAN, BadActs, MDP) are weak in this setting. The authors propose BandAid, a plug-in patch that uses suspicious samples flagged by a base defense as noisy poison labels, combines them with a small clean validation set, and trains a classifier to improve poison detection/mitigation. The paper evaluates the method across multiple datasets, attacks, poisoning intensities, and FRR settings, and reports strong empirical gains in many cases.

**Audience:**

Yes

**Audience Explanation:**

Yes. The paper is relevant to readers interested in NLP robustness, model security, and inference-time defenses. Clean-label backdoors are a realistic and difficult threat model, and the paper fills an important evaluation gap by showing that methods that look good on dirty-label attacks can fail badly here. The proposed patch is also practical and easy to understand, which increases the chance that other researchers can test or adopt it. I think the audience will find both the stress-test results and the BandAid design useful

**Broader Impact Concerns:**

The work is defensive in intent, and I do not see major ethical concerns beyond standard dual-use concerns in security research.

**Claims And Evidence:**

Yes

**Claims Explanation:**

The paper provides empirical evidence for its main claims. The evaluation setup is broad and directly aligned with the paper’s thesis: three SOTA inference-time defenses are stress-tested under clean-label attacks with multiple attack families, datasets, poisoning rates, and training intensities. The results consistently show that the base defenses are brittle in clean-label settings and that BandAid usually improves attack mitigation and often improves clean accuracy as well. The paper also supports its interpretation with useful analyses: TP/FP counts from the base defenses, FRR trade-offs, and the warmup/efficiency experiment showing how many suspicious samples are needed before BandAid becomes effective. This makes the core argument convincing, not just the headline numbers.

**Requested Changes:**

1. The current framing presents BandAid as a broad plug-in patch, but the method is strongly dependent on base-defense quality and is evaluated only in a limited setting (RoBERTa-base text classification). The paper should narrow the claim to match the evidence and remove or qualify “universal” throughout unless additional experiments are added.
2. The paper needs ablations on: (a) classifier architecture (for example, smaller model, frozen encoder + linear head), (b) amount of clean validation data, (c) sensitivity to noisy pseudo-labels in the flagged set, and (d) class imbalance between flagged and clean samples. These are necessary to show that the proposed mechanism is stable and not tied to one specific training setup.
3. The evaluation should clearly report method-by-method threshold settings, FRR relaxations, non-convergence/timeouts, and any exclusions. This should be in the main paper, not only scattered in appendix details. Without this, it is difficult to judge whether the comparisons are fair.

---

### Review · Reviewer_LwpF · 2026-03-02

**Summary Of Contributions:**

This paper makes two main contributions. First, it systematically evaluates three state-of-the-art inference-time backdoor defenses (DAN, BadActs, and MDP) under clean-label attack settings and demonstrates that they largely fail. Second, it proposes BandAid, a plug-in module that collects suspicious samples flagged by an existing defense, treats them as "poison" labels, and fine-tunes a lightweight RoBERTa-base classifier together with a small clean validation set to re-classify incoming test samples. The core reframing is turning anomaly detection into discriminative classification. Across 102 evaluation cases, BandAid improves upon the base defense in 99 of them, reducing ASR by up to 99.8% and improving CACC by 7.0% on average.
The experiments span three datasets (SST-2, AG News, Blog), six attack strategies (Addsent, SynBkd, LLMBkd Bible/Gen-Z, AttrBkd Bible/Tweets), and sweep across poisoning rates, training intensities, and FRR thresholds. The overall experimental scope is substantial.

Key Strengths:

The problem is well-motivated; clean-label attacks are a genuine blind spot for existing defenses.

BandAid is simple and empirically effective.

Evaluation covers multiple dimensions including poisoning rate, training intensity, and FRR threshold.

Key Weaknesses:

BandAid is not a standalone defense and is entirely dependent on the quality of the upstream detector.

The paper lacks theoretical grounding or ablation studies to explain why fine-tuning on noisy suspicious samples works so well.

Some anomalous results (e.g., MDP + BandAid completely failing at 1% PR on AttrBkd) are not adequately explained.

**Audience:**

Yes

**Audience Explanation:**

Defending against clean-label backdoor attacks is an open and practically important problem. The plug-in design of BandAid is appealing from an engineering standpoint, and the reframing of "noisy anomaly detection outputs" as "labeled training data for a discriminative classifier" is a useful conceptual contribution. Even setting aside BandAid itself, the thorough stress-testing of three SOTA defenses under clean-label conditions has value as a standalone empirical finding. The community would benefit from knowing that DAN, BadActs, and MDP all fail in non-trivial ways in this setting.

**Broader Impact Concerns:**

The paper does not include a Broader Impact Statement. Given that the work is primarily defensive in nature, the ethical concerns are limited. However, one assumption underlying BandAid deserves mention in the Limitations section (currently Section 8): BandAid relies on a small clean validation set to anchor the classifier. In a more adversarial deployment setting where the attacker has partial access to the validation pipeline, this assumption could be violated. The current Limitations section does not discuss the security of BandAid's own training data, which is a meaningful gap.

**Claims And Evidence:**

Yes

**Claims Explanation:**

The central claim that"BandAid improves upon the base defense in 99 out of 102 cases" is supported by the data across Tables 2 and 3, which span three datasets and six attacks. However, several specific issues warrant attention.

First, the paper never clearly defines how the "102 cases" are counted. The dimensionality (3 defenses × 3 datasets × 6 attacks = 54, or including FRR/intensity variants) is ambiguous. Readers should not have to reverse-engineer this from the tables. The authors should state explicitly which combinations constitute the 102 cases, and identify which three failed.

Second, BandAid's gains on the Blog dataset are substantially weaker than on SST-2 and AG News. For instance, DAN + BandAid against AttrBkd (Tweets) achieves ASR = 0.291 on Blog (Table 2), compared to 0.004 on SST-2 and 0.013 on AG News. Blog has a base CACC of only 55.2% (Table 1) and is a 3-class authorship dataset with different distributional properties. This discrepancy deserves dedicated analysis rather than being folded into aggregate statistics.

Third, Table 7 (also duplicated as Table 17 in the appendix) shows MDP + BandAid achieving ASR = 0.960 on AttrBkd (Tweets) at 1% PR — identical to the undefended model. The authors explain in Section 5.2 that "MDP identifies only seven true poison samples, which is insufficient." However, Table 7 shows TP = 163 at FRR = 3% for MDP on the same attack. It is unclear whether "seven" refers to a different poisoning rate, FRR setting, or dataset split. This inconsistency needs to be resolved.

Fourth, the "7.0% average CACC improvement" is heavily influenced by MDP's catastrophically low CACC on AG News (ranging from 0.277 to 0.575 in Table 3). Including these as a baseline for measuring BandAid's improvement inflates the average significantly. The authors should either report this metric separately for cases where the base defense does not cause catastrophic accuracy degradation, or explicitly acknowledge that much of the gain is attributable to correcting MDP's failures rather than a general improvement.

**Requested Changes:**

The following are ordered by importance:

Critical:

BandAid lacks theoretical or mechanistic explanation. The paper argues qualitatively that discriminative classification is more powerful than anomaly detection, but provides no analysis of why a classifier trained on noisy, mixed-label data outperforms the defense that generated that data. A key ablation is missing: what happens if the "poison" training samples for BandAid are replaced with random negatives, rather than the true suspicious samples flagged by the defense? This would isolate how much of BandAid's effectiveness depends on the TP content versus simply having a fine-tuned binary classifier. Similarly, the ratio of TP to FP in BandAid's training set varies widely across settings (e.g., DAN on Addsent: 85 TP / 124 FP vs. DAN on LLMBkd Bible: 1743 TP / 122 FP, from Table 4), yet BandAid performs well in both cases. Understanding why the model is robust to this noise would significantly strengthen the paper.

The "102 cases" statistic must be precisely defined. Provide a table or appendix entry enumerating all 102 cases, along with identification of the three where BandAid fails and a corresponding analysis. As currently written, this central claim cannot be independently verified.
The Blog dataset requires more careful discussion. A classifier operating at ~55% accuracy on a 3-way classification task is arguably unreliable to begin with, and this affects the interpretability of both ASR and CACC results. The comparatively weak BandAid gains on Blog (e.g., DAN + BandAid ASR = 0.291 for AttrBkd Tweets, vs. near-zero on other datasets) need explanation. Is this a data quality issue, a class imbalance artifact, or a fundamental limitation of the approach on harder classification tasks?

MDP's catastrophic CACC degradation on AG News and Blog is a major finding that deserves dedicated analysis. MDP achieves CACC as low as 0.277 on AG News (Table 3, LLMBkd Bible) and 0.324 on Blog (AttrBkd Tweets). The current text dismisses this with "MDP is inadequate across all evaluated attacks and datasets." This failure mode — where a defense actively harms clean accuracy more than the attack itself does — is important for the community to understand. Is this a consequence of MDP's random masking disrupting multi-class prompt-based inference? Does it reflect a sensitivity to dataset size?

Non-critical:

Figure 4 shows BandAid's learning curve only when built on top of DAN. Given that BadActs and MDP provide different TP/FP mixtures (as shown in Table 4), their corresponding learning curves could look quite different. At minimum, the authors should comment on whether BandAid's sample efficiency is consistent across base defenses.

BadActs failing to converge for Addsent and SynBkd (shown as "−" in Table 2) is noted only in a footnote. Given that convergence failure after 24 hours is a significant practical limitation, this deserves a sentence or two in the main text.

The claim that "BandAid aligns with the principle behind backdoor attacks themselves" is conceptually interesting but underdeveloped. Unpacking this analogy more carefully, specifically, that a small number of consistently patterned samples can dominate a fine-tuning signal, would help readers appreciate why even a noisy set of suspicious samples is sufficient for BandAid to work.

In Table 8, DAN + BandAid on Blog under no attack shows CACC = 0.523 versus DAN alone at 0.536. The authors' claim that BandAid "maintains comparable or slightly improved clean accuracy" is imprecise here. The drop is small, but it is a drop, and the language should reflect this honestly.

---

### Review · Reviewer_zMKs · 2026-03-13

**Summary Of Contributions:**

BandAid proposes a universal plug-in module to enhance inference-time backdoor defenses in natural language processing. The authors observe that current state-of-the-art defenses like DAN, BadActs, and MDP struggle to distinguish between clean and poisoned samples in clean-label scenarios. To bridge this gap, the framework uses a small set of clean validation data to refine the decision boundaries of existing defenses. The methodology is evaluated across multiple datasets including IMDb, Blog, AgNews, and Tweets, covering both insertion-based and stylistic paraphrase-based attacks. The authors show that BandAid significantly reduces attack success rates in 99 out of 102 cases, with some reductions reaching 99.8% while simultaneously improving clean data accuracy by 7.0% on average.

**Audience:**

Yes

**Audience Explanation:**

The findings of this paper would interest a portion of the TMLR audience, especially those focused on machine learning security in NLP domain. As backdoor attacks become more sophisticated and move toward stealthy clean-label methods, developing modular tools like BandAid that can be integrated into existing pipelines is a high-priority area.

**Broader Impact Concerns:**

This paper does not provide the broader impact statement. However, there are no major broader impact concerns in this paper, as it provides a defensive tool that improves the security and reliability of NLP models used in the real world.

**Claims And Evidence:**

Yes

**Claims Explanation:**

The claims in the submission are supported by accurate and clear evidence. The experimental evaluation is comprehensive, covering various NLP tasks, diverse backdoor triggers, and different poisoning rates. The comparison against baselines convincingly demonstrates that while current defenses fail in clean-label settings, the addition of BandAid drops success rates to near-zero. The use of False Rejection Rate thresholds and True Positive rates provides a transparent view of the security-utility trade-offs.

**Requested Changes:**

1. The authors should provide more detail on the inference-time latency added by the plug-in to assess its practical feasibility in production environments.
2. It would also be beneficial to include an analysis of how the size and diversity of the clean validation set impact the final decision boundary.
3. The authors should discuss potential adaptive attackers who might attempt to bypass the defense by crafting samples that specifically mimic the distribution of the clean validation set.

---

### Decision · Action_Editor_t8UT · 2026-06-07

**Recommendation:** Reject

**Audience:**

Yes

**Audience Explanation:**

The paper addresses machine learning safety issue which is quite relevant to the machine learning community.

**Claims And Evidence:**

No

**Claims Explanation:**

Reviewers considered the paper addressing a well-motivated problem, and the solution is simple yet seem effective. Multiple questions regarding empirical evaluation have been raised, including scope of backbones, missing ablation studies, inconsistency in reported results. Unfortunately, no rebuttal were provided, and reviewers remained unconvinced. The paper is not yet ready for publication in its current form.

**Resubmission Of Major Revision:**

The authors may consider submitting a major revision at a later time.